# A Systematic Study of Performance Disparities in Multilingual Task-Oriented Dialogue Systems

**Songbo Hu**[1]    **Han Zhou**[1]    **Zhangdie Yuan**[2]    **Milan Gritta**[3]
**Guchun Zhang**[3]    **Ignacio Iacobacci**[3]    **Anna Korhonen**[1]    **Ivan Vulić**[1]

[1]Language Technology Lab, University of Cambridge, UK
[2]Department of Computer Science and Technology, University of Cambridge, UK
[3]Huawei Noah's Ark Lab, London, UK
[1,2]{sh2091,hz416,zy317,alk23,iv250}@cam.ac.uk
[3]{milan.gritta,guchun.zhang,ignacio.iacobacci}@huawei.com

## Abstract

Achieving robust language technologies that can perform well across the world's many languages is a central goal of multilingual NLP. In this work, we take stock of and empirically analyse *task performance disparities* that exist between multilingual task-oriented dialogue (ToD) systems. We first define new quantitative measures of *absolute* and *relative equivalence* in system performance, capturing disparities across languages and within individual languages. Through a series of controlled experiments, we demonstrate that *performance disparities* depend on a number of factors: the nature of the ToD task at hand, the underlying pretrained language model, the target language, and the amount of ToD annotated data. We empirically prove the existence of the *adaptation bias* and *intrinsic biases* in current ToD systems: e.g., ToD systems trained for Arabic or Turkish using annotated ToD data fully parallel to English ToD data still exhibit diminished ToD task performance. Beyond providing a series of insights into the *performance disparities* of ToD systems in different languages, our analyses offer practical tips on how to approach ToD data collection and system development for new languages.

## 1  Introduction

The aim of task-oriented dialogue (ToD) (Gupta et al., 2006; Tür et al., 2010; Young, 2010) is to model the interaction between a human user and a system agent with the goal of accomplishing specific, well-defined tasks. To date, ToD technology has proven useful for many sectors, ranging from the hospitality industry (Henderson et al., 2014, 2019) to healthcare (Laranjo et al., 2018), online shopping (Yan et al., 2017), banking (Altinok, 2018), and travel (Raux et al., 2005; El Asri et al., 2017), among others. These systems provide users with access to state-of-the-art services and they catalyse technological expansion.

The development of ToD systems requires large-scale, in-domain datasets to exploit the potential of deep learning-based components to effectively handle complex dialogue patterns (Budzianowski et al., 2018; Lin et al., 2021b; Hu et al., 2023). The creation of datasets for new domains and languages is challenging (Shah et al., 2018; Larson and Leach, 2022): it requires expertise and substantial time and financial investment that typically exceeds the requirements of other NLP tasks (Casanueva et al., 2022). Therefore, the progress in ToD is still largely confined to a small number of high-resource languages (Razumovskaia et al., 2022).

On the other hand, recent advances in multilingual pretrained language models (mPLMs) (Devlin et al., 2019; Conneau et al., 2020a; Xue et al., 2021a) have conceptually enabled cross-lingual transfer between any two or more languages seen at pretraining (Wu and Dredze, 2019; Conneau et al., 2020b; Hu et al., 2020), or even to unseen languages (Ansell et al., 2021). These models offer a promising basis for developing ToD systems for previously unsupported languages, obviating the need for expensive, language-specific data acquisition. However, mPLMs do not equally benefit all languages: previous studies focused on standard NLP tasks (Hu et al., 2020; Lauscher et al., 2020) have revealed that they exhibit unequal performance across languages, with particularly adverse effects observed for low-resource languages.

The scarcity or complete absence of high-quality in-domain ToD data (i.e., *adaptation bias*; Bommasani et al. 2021) and the compounding under-representation of target languages inherent to mPLMs (i.e., *intrinsic bias*; Bommasani et al. 2021) are likely to collectively contribute to the exacerbation of *performance disparities* in multilingual ToD systems. These disparities, specifically *error rate disparities* (Barocas et al., 2019), may result in *extrinsic harms* for downstream applications (Galliers and Spärck Jones, 1993), affecting the utility,

experience, and satisfaction of system users. They may also deprive non-English speakers of *equal opportunities*, hampering their ability to benefit from the rapid and ever-growing advancements in language technology.

Aligned with the broader goal (Council of European Union, 2018) of developing multilingually inclusive NLP, this work offers a first systematic analysis of the *performance disparities* exhibited by multilingual ToD systems.[1] In §3, we first propose two measures for studying performance disparity, namely the *absolute* and *relative equivalence* in system performance. The former evaluates the disparities in task performance across languages, while the latter measures the achieved performance compared to its potential 'upper bound' performance within the same language.

We apply these measures to study performance disparities in a recently developed MULTI3WOZ resource (Hu et al., 2023), a large-scale multilingual multi-domain multi-parallel ToD dataset.

In particular, relying on MULTI3WOZ, we investigate the following research questions:

*(RQ1) Given the recent progress in mPLMs, Machine Translation, and cross-lingual transfer, is language-specific data still necessary for the development of a* ToD *system for a new language?*

*(RQ2) Given access to the same mPLMs, equivalent amounts of high-quality in-language training data, and a similar development approach as that used to create an English* ToD *dataset, is it possible to develop a* ToD *system for a new language that achieves near-English performance?*

*(RQ3) How much training data is required in a new language to achieve performance comparable to a* ToD *system trained with an equivalent amount of in-domain, in-language data as in English?*

*(RQ4) Which data collection strategy maximises system performance across metrics while minimising the amount of annotation required? Such a strategy could optimise the cost-efficiency of annotation for a new language.*[2]

Our empirical findings highlight *performance*

*disparities* between English and other languages even when the development approaches are similar. Such disparities can be alleviated through in-language data collection, with the necessary data volume increases as the task complexity rises. Additionally, we demonstrate the feasibility and benefits of strategic annotation budget allocation in multilingual dialogue collection, showing how model performance can be enhanced without additional annotation costs.

We hope that the key findings of this study, beyond empirically validating the current *performance disparities* in multilingual ToD systems, will also serve as practical guidelines for ToD system developers, contributing to the long-term goal of constructing robust and inclusive ToD systems for a much larger number of languages, aligning with the recent initiatives in MT (Siddhant et al., 2022; NLLB Team et al., 2022) and other NLP tasks (Søgaard, 2022; Ruder et al., 2023). Our data and code are available at: `https://github.com/cambridgeltl/multi3woz/tree/analysis`.

## 2   Related Work

**Multilingual ToD Systems and Datasets.** Previous work has shown that fine-tuning methods are competitive or even outperform in-context learning with large language models (LLMs) for cross-lingual transfer (Asai et al., 2023) and ToD systems (Hudeček and Dušek, 2023; Heck et al., 2023). This applies even with smaller models and when the number of training examples is limited. Therefore, state-of-the-art multilingual ToD systems are typically developed by fine-tuning mPLMs with domain-specific datasets.

Developing ToD systems without the disadvantage of data scarcity[3] necessitates a dataset that has the following properties (Hu et al., 2023): (i) **large-scale data**, containing a sufficient volume of (both training and test) data for in-depth comparative cross-language analyses, (ii) **comprehensive task-specific annotations**, supporting the training of multiple ToD tasks, (iii) **natural-sounding conversations**, ensuring natural flow within the dialogues and avoiding artificial performance inflation (Majewska et al., 2023), and (iv) **wide language coverage**, enabling controlled experiments

---

[1]In this study, we use the terms *performance disparity* and *error rate disparity* interchangeably. We acknowledge that in real-world production settings, there are various evaluation dimensions to assess system performance, with error rate, while critical, being only one of them.

[2]A related experimental study conducted by Debnath et al. (2021) on multilingual QA systems emphasises taking the most advantage of existing resources rather than expanding the system to languages that have not been previously supported.

[3]The term 'disadvantage of data scarcity' in this context specifically refers to the limited quantity and/or quality of in-domain in-language training data for multilingual ToD systems. This data scarcity issue is a major source of the adaptive bias mentioned earlier.

and comparison across a representative set of languages and in different linguistic contexts.

The majority of the existing ToD datasets fail to simultaneously satisfy all the aforementioned properties. For example, most existing multilingual ToD datasets (Upadhyay et al., 2018; Schuster et al., 2019; Dao et al., 2021; Moghe et al., 2022; Majewska et al., 2023, *inter alia*) have been designed to support a single component within a ToD system, typically Natural Language Understanding (NLU), failing to fulfil property (ii). Recently, several datasets have emerged that support multiple dialogue tasks, but again none of these satisfy all the properties mentioned above. For instance, some datasets such as Multi2WOZ (Hung et al., 2022) do not provide any training data at all, while others like AllWOZ (Zuo et al., 2021) only provide a minimal training set of post-edited machine-translated dialogues. Datasets such as GlobalWOZ (Ding et al., 2022) rely solely on automatically created machine-translated training data, failing to fulfil properties (i) and (iii). BiToD (Lin et al., 2021b) is reasonably large and features coherent dialogues, but it spans only two, highest-resourced languages, failing (iv).

Our study is enabled by a new multilingual ToD dataset that overcomes the limitations of earlier datasets: MULTI3WOZ– the first large-scale, multilingual multi-domain multi-parallel ToD dataset that has all of the properties discussed above and can, therefore, serve as a comprehensive benchmark and a departure point for our analysis. For completeness, we provide a summary description of MULTI3WOZ in §3.3.

## 3 Methodology

### 3.1 Measuring Performance of ToD Systems

This study investigates the *performance disparities* observed in ToD systems across different languages. Ideally, to be inclusive and widely useful, multilingual ToD systems should attain comparable performance for all the languages for which NLP can be developed. The development of strongly multilingual systems that can deal with diverse linguistic patterns is also important for robustness of ToD systems. However, measuring the full generality of system performance is challenging: it requires a comprehensive procedure involving e.g., a large-scale user study and the apparatus of measurement theory (Barocas et al., 2019). This is only attainable via collaborative efforts of the research

community as it involves addressing long-standing challenges in dialogue system evaluation (Mehri et al., 2022). Our study focuses on a more realistic task: measurement of *performance disparities* using established automatic evaluation metrics as proxies to measure users' judgement of the system. This approach aligns with similar methodologies adopted in related works that assess system performance on other NLP tasks (Blasi et al., 2022; Khanuja et al., 2023, *inter alia*).

### 3.2 Notion of Equivalence in Performance

**Preliminaries and Notation.** The development of a ToD system involves training a dialogue model, denoted as $P(\cdot)$, using a task-specific dialogue dataset $\mathbb{D}$. The system's performance is then evaluated using an automatic evaluation metric $\mathrm{M}(\cdot)$. In a multilingual setup, the dialogue model $P(\cdot)$ is commonly implemented using an mPLM. The training procedure involves cross-lingual transfer training, utilising a dataset $\mathbb{D}^{\mathrm{SRC}}$ in a typically high-resource source language, and a dataset $\mathbb{D}^{\mathrm{TGT}}$ in a low-resource target language. The following discussion is made under an ideal assumption that the datasets $\mathbb{D}^{\mathrm{SRC}}$ and $\mathbb{D}^{\mathrm{TGT}}$ are of equal size and quality, and cover exactly the same conversational flows and information types (which is provided by MULTI3WOZ). We assume that the system developed in the target language is not affected by in-domain data scarcity, domain shifts, or different in-domain distributions.

**Absolute $\theta$-Equivalence.** We train a system, denoted as $P^{\mathrm{SRC}}(\cdot)$, on the source language dataset $\mathbb{D}^{\mathrm{SRC}}$ and evaluate its performance using the metric $\mathrm{M}(P^{\mathrm{SRC}}(\cdot))$. Similarly, we train a system for the target language, denoted as $P^{\mathrm{TGT}}(\cdot)$, using the target language dataset $\mathbb{D}^{\mathrm{TGT}}$. Subsequently, we assess the performance of the target language system using the metric $\mathrm{M}(\cdot)$. To quantify performance disparities in systems development across different languages, we introduce the concept of absolute $\theta$-equivalence. This concept represents a performance threshold of a system developed in the target language, which is compared against the fully supervised system in the source language. We define that two systems achieve absolute $\theta$-equivalence iff $\mathrm{M}(P^{\mathrm{TGT}}(\cdot)) \geq \theta \cdot \mathrm{M}(P^{\mathrm{SRC}}(\cdot))$, where $\theta \in [0,1]$.[4] For instance, the ToD system

---

[4]The relationship between evaluation metrics and the actual benefits experienced by users can be complex and nonlinear (Blasi et al., 2022). For instance, in the domain of

developed for French may demonstrate a performance level of 0.95-equivalence, indicating a substantial degree of equivalence in comparison to the system in the source language. However, it may not achieve the performance levels of 0.99-equivalence or perfect 1-equivalence. Unless noted otherwise, we use a specific example $\theta$ value of 0.95 to facilitate the comparison in this study.

To ensure an equal performance in each language, the ideal scenario would involve two systems achieving absolute 1-equivalence. However, our experimental results in §4 *(RQ2)* will reveal a significant disparity between systems developed for English and other languages, even when trained on an equal amount of dialogue data. This discrepancy can be attributed to the *inherent bias* of the underlying mPLM, which cannot be easily mitigated, especially when it comes to low-resource languages. To gain deeper insights into performance disparities, we also define relative $\theta$-equivalence.

**Relative $\theta$-Equivalence.** It is challenging to collect a large in-domain, in-language training dataset for ToD. Consequently, an in-language training dataset $\mathbb{D}_{few}^{\text{TGT}}$ is often considerably smaller in size compared to $\mathbb{D}^{\text{TGT}}$ and $\mathbb{D}^{\text{SRC}}$. Assuming that the full $\mathbb{D}^{\text{TGT}}$ is available, we train a system, denoted as $P^{\text{TGT}}(\cdot)$, for the target language. Additionally, we train another system, $P_{few}^{\text{TGT}}(\cdot)$, using either the limited target language dataset $\mathbb{D}_{few}^{\text{TGT}}$ or a combination of $\mathbb{D}^{\text{SRC}}$ and $\mathbb{D}_{few}^{\text{TGT}}$ through cross-lingual training. To quantify the performance disparities arising from the scarcity of in-language training data for a target language, we employ the term relative $\theta$-equivalence. This concept represents a performance threshold for a system developed in the target language, which is compared against the system trained with a full dataset in the same target language. We define that the two systems achieve relative $\theta$-equivalence iff the metric $\text{M}(P_{few}^{\text{TGT}}(\cdot)) \geq \theta \cdot \text{M}(P^{\text{TGT}}(\cdot))$, where $\theta \in [0,1]$.

Relative $\theta$-equivalence threshold for higher $\theta$ values might be achievable by enlarging the size of $\mathbb{D}_{few}^{\text{TGT}}$ without changing the underlying mPLM. By focusing on strategically expanding the target language dataset, we can analyse the impact of *adaptive bias* in ToD data without explicitly modelling the compounding intrinsic bias of the model.

---

machine-assisted human translation, the correlation between machine translation accuracy and productivity gain can exhibit intricate patterns (Sanchez-Torron and Koehn, 2016). In this study, we simplify the relationship between the chosen evaluation metric and users' judgement of the system.

## 3.3 MULTI3WOZ: A Quick Recap

MULTI3WOZ (Hu et al., 2023) is a new large-scale multilingual, multi-domain, and multi-parallel ToD dataset that, unlike previous datasets, fully meets the criteria mentioned in §2 for our investigation: it provides training, development, and test sets in both the source and target languages (i.e., $\mathbb{D}^{\text{SRC}}$ and $\mathbb{D}^{\text{TGT}}$) that are not only of equal in size and created using the same protocols by native speakers of the respective languages, but also possess comparable quality through covering the same conversation flows via multi-parallel dialogues.

This dataset $\mathbb{D}$ comprises a total of 36,640 (4×9,160) parallel yet linguistically and culturally adapted dialogues in four languages: Arabic ($\mathbb{D}^{\text{ARA}}$; Afro-Asiatic), English ($\mathbb{D}^{\text{ENG}}$; Indo-European), French ($\mathbb{D}^{\text{FRA}}$; Indo-European), and Turkish ($\mathbb{D}^{\text{TUR}}$; Turkic), constructed by leveraging the well-established multi-domain English Multi-WoZ dataset (Budzianowski et al., 2018), particularly its cleaned version 2.3 (Han et al., 2021). In this study, we consider the English dialogues $\mathbb{D}^{\text{ENG}}$ as the source dataset $\mathbb{D}^{\text{SRC}}$. Additionally, among other languages, each dataset $\mathbb{D}^{\text{ARA}}$, $\mathbb{D}^{\text{FRA}}$, and $\mathbb{D}^{\text{TUR}}$ is treated as a target dataset $\mathbb{D}^{\text{TGT}}$. Each target dataset is generated by adapting a recent bottom-up outline-based approach introduced by Majewska et al. (2023) to address the limitations of translation-based design. For further technical details concerning MULTI3WOZ, we refer the reader to the original work (Hu et al., 2023).

## 4 Experimental Setup

Our experiments involve three standard ToD tasks: Natural Language Understanding (NLU), Dialogue State Tracking (DST), and Natural Language Generation (NLG). We briefly recap each task along with its experimental setup.[5]

**NLU.** This task is commonly decomposed into two well-established subtasks: intent detection (ID) and slot labeling (SL). In ID, the objective is to classify the user's utterance and to determine the presence of a specific domain-intent pair from a predefined set of intents in the ontology. It is treated as a multi-class classification task. SL is a sequence tagging task that identifies the presence of a value and its corresponding slot within the utterance.[6]

---

[5]We elaborate on all the other details in Appendix A.

[6]For example, *Restaurant-Inform* is the domain-intent pair for the utterance *There will be 5 of us and 19:45 would be great.* The value *5* corresponds to the slot *number_of_people*,

We evaluate ID and SL methods implemented atop XLM-R$_{base}$ and XLM-R$_{large}$ (Conneau et al., 2020a). The evaluation is conducted by measuring the accuracy of correctly identifying the presence of all domain-intent pairs, as well as its F1 score. For SL, we employ the commonly used BIO labelling scheme to annotate each token in the user's utterance. The evaluation of SL is based on F1 score, precision, and recall in accurately identifying each slot value within the utterance.

**DST.** Our DST models are based on T5DST (Lin et al., 2021a), a strong DST baseline which transforms DST into a QA task by incorporating slot descriptions. For implementation, we utilise multiple multilingual sequence-to-sequence models, including mT5$_{small}$, mT5$_{large}$ (Xue et al., 2021b), Flan-T5$_{small}$, and Flan-T5$_{large}$ (Chung et al., 2022). To evaluate our approach, we follow the standard MultiWOZ preprocessing and evaluation setups (Wu et al., 2019), excluding the 'hospital' and 'police' domains due to the absence of test dialogues in these domains. We report Joint Goal Accuracy (JGA), Turn Accuracy, and Joint F1.[7]

**NLG.** We approach the NLG task as a sequence-to-sequence problem, again supported by mT5$_{small}$, mT5$_{large}$, Flan-T5$_{small}$, and Flan-T5$_{large}$. Our approach involves taking the concatenation of the two preceding historical utterances and the linearised 'oracle' dialogue act (e.g., *[inform][restaurant]([price range][expensive],[area][center]*) as input to generate a system response. Following MultiWOZ conventions, we evaluate with the corpus BLEU score (Papineni et al., 2002). However, we evaluate lexicalised utterances without performing delexicalisation. We also report ROUGE-L (Lin, 2004) and METEOR (Banerjee and Lavie, 2005).

**One More Thing...** We emphasise the wide applicability of our proposed notions of $\theta$-equivalence, as they are not restricted to any specific metric, model, or dataset. They are task-agnostic notions applicable to a diverse range of models beyond those mentioned in ToD tasks. We propose using them as a tool to measure *performance dispari-*

---

and the value *19:45* corresponds to the slot *time_of_booking*.

[7]The JGA measure represents the proportion of dialogue turns in the dataset where all slots have been correctly filled with their ground truth values. For assessing turn accuracy, each (domain, slot, value) triplet is compared against its corresponding ground truth label. Furthermore, the joint F1 score is computed as the macro-averaged F1 score across all these slots, considering all the turns present in the dataset.

*ties* across different domains and multilingual NLP tasks, thereby promoting the development of inclusive and linguistically robust systems.

# 5 Results and Discussion

Centred around the definitions of $\theta$-equivalence, our experiments are focused on answering research questions from §1. Our primary objective is to identify, quantify, and mitigate performance disparities within multilingual ToD systems. To solidify our empirical findings, we present supplementary experimental results in Appendix B; they offer additional support to our main claims as well as auxiliary findings.

To answer *RQ1*, we conduct evaluations of mPLMs in three standard scenarios: fully supervised, zero-shot cross-lingual transfer, and translate train. Our empirical results highlight the necessities of language-specific data in the development of a ToD system for a new language. In response to *RQ2*, our subsequent analysis highlights performance disparities among task-specific dialogue models, even when trained with equivalent amounts of data. We further examine model performances in both few-shot and 'many'-shot settings, shedding light on the correlations between task complexity and the required amount of data, thus addressing *RQ3*. Finally, we investigate the advantages of collecting a dataset that maximises lexical coverage, achieving improved cost-efficiency of annotation, answering *RQ4*.

**(RQ1) Supervised, Translation, and Zero-shot.** To answer *RQ1*, we examine the performance of individual modules in multilingual ToD systems under three scenarios: (1) the ideal situation, where a multi-parallel dataset in the target language $\mathbb{D}^{TGT}$ is available in equal size and quality as English, (2) the real life situation for most languages, where there is a complete absence of in-domain in-language training data, and (3) the standard translate-train situation for languages that have a reliable machine translation (MT) system but lack $\mathbb{D}^{TGT}$. In the latter case, we employ an MT system to generate a dataset $\mathbb{D}^{TGT}_{mt}$ in the target language from $\mathbb{D}^{ENG}$.

In scenario (1), we adopt a fully supervised approach for training and evaluating individual languages separately. For example, we train and evaluate systems $P^{ARA}(\cdot)$ exclusively on the Arabic dataset $\mathbb{D}^{ARA}$. In scenario (2), we perform zero-shot cross-lingual transfer evaluation for each tar-

| Language | Intent Detection | | Slot Labelling | | | Dialogue State Tracking | | | Natural Language Generation | | |
|---|---|---|---|---|---|---|---|---|---|---|---|
| | Accuracy | F1 | Precision | Recall | F1 | JGA | Turn Acc. | F1 | BLEU | ROUGE | METEOR |
| **Fully Supervised** | | | | | | | | | | | |
| ENG | $93.2_{92.0}$ | $96.1_{95.3}$ | $94.6_{93.6}$ | $95.7_{96.0}$ | $95.1_{94.8}$ | $57.2_{59.8}$ | $97.7_{97.9}$ | $92.5_{93.5}$ | $20.1_{20.8}$ | $47.3_{48.4}$ | $42.9_{44.1}$ |
| ARA | $92.7_{92.1}$ | $95.0_{94.6}$ | $42.4_{42.2}$ | $48.5_{48.1}$ | $45.2_{45.0}$ | $42.0_{47.9}$ | $96.4_{96.9}$ | $88.0_{89.4}$ | $6.8_{17.9}$ | $0.8_{15.0}$ | $19.4_{36.0}$ |
| FRA | $89.2_{88.6}$ | $93.0_{92.6}$ | $76.9_{77.1}$ | $79.2_{79.1}$ | $78.0_{78.1}$ | $47.6_{49.7}$ | $96.8_{97.0}$ | $89.4_{90.1}$ | $12.9_{13.9}$ | $39.6_{40.9}$ | $33.8_{35.2}$ |
| TUR | $92.2_{91.5}$ | $95.0_{94.4}$ | $76.9_{77.1}$ | $87.6_{87.3}$ | $87.1_{86.9}$ | $50.5_{52.9}$ | $97.1_{97.3}$ | $90.5_{91.2}$ | $5.5_{24.2}$ | $24.7_{53.7}$ | $22.5_{48.6}$ |
| **Zero-shot Cross-lingual Transfer** | | | | | | | | | | | |
| ARA | $82.1_{65.7}$ | $88.2_{74.8}$ | $27.4_{17.2}$ | $31.2_{27.7}$ | $29.2_{21.2}$ | $1.9_{1.5}$ | $82.5_{80.7}$ | $17.0_{5.8}$ | $0.2_{0.2}$ | $2.5_{2.1}$ | $2.4_{2.0}$ |
| FRA | $83.9_{77.0}$ | $89.8_{85.0}$ | $58.5_{49.1}$ | $61.2_{62.4}$ | $59.8_{54.9}$ | $5.5_{3.7}$ | $86.6_{85.1}$ | $40.1_{32.8}$ | $0.5_{0.4}$ | $4.2_{4.7}$ | $6.1_{5.9}$ |
| TUR | $87.0_{74.9}$ | $91.4_{81.7}$ | $68.1_{48.5}$ | $74.7_{66.6}$ | $71.2_{56.2}$ | $3.5_{1.3}$ | $85.2_{82.1}$ | $34.4_{15.2}$ | $0.3_{0.4}$ | $3.7_{4.4}$ | $6.1_{5.8}$ |
| **Translate Train** | | | | | | | | | | | |
| ARA | $72.0_{67.3}$ | $81.9_{78.9}$ | $0_0$ | $0_0$ | $0_0$ | $9.2_{32.4}$ | $89.1_{94.2}$ | $52.7_{79.9}$ | $1.1_{1.2}$ | $6.3_{6.7}$ | $7.4_{7.6}$ |
| FRA | $66.2_{63.4}$ | $77.4_{74.9}$ | $0_0$ | $0_0$ | $0_0$ | $10.4_{9.8}$ | $90.6_{90.6}$ | $60.0_{58.7}$ | $2.6_{3.2}$ | $20.4_{23.2}$ | $15.1_{17.8}$ |
| TUR | $71.2_{66.5}$ | $82.2_{78.6}$ | $0_0$ | $0_0$ | $0_0$ | $10.5_{32.9}$ | $90.5_{94.3}$ | $60.4_{79.7}$ | $1.0_{1.0}$ | $16.9_{17.4}$ | $12.7_{13.0}$ |

Table 1: (1) Fully supervised, (2) zero-shot cross-lingual transfer from English $\mathbb{D}^{\text{ENG}}$, and (3) translate train from GlobalWOZ 'F&E' dataset $\mathbb{D}^{\text{TGT}}_{mt}$ for ID, SL, DST, and NLG tasks on MULTI3WOZ. Performance evaluations were conducted for two distinct model categories: large models, namely XLM-R$_{\text{large}}$ and mT5$_{\text{large}}$, and small models, specifically XLM-R$_{\text{base}}$ and mT5$_{\text{small}}$. The reported results follow the format of "large model$_{\text{small model}}$". For example, the entry $93.2_{92.0}$ denotes XLM-R$_{\text{large}}$ achieves 93.2 accuracy in English ID whereas XLM-R$_{\text{base}}$ achieves 92.0.

get language. To achieve this, we train an English system $P^{\text{ENG}}(\cdot)$ using the $\mathbb{D}^{\text{ENG}}$ dataset, and subsequently assess its performance on the target language datasets, namely $\mathbb{D}^{\text{ARA}}$, $\mathbb{D}^{\text{FRA}}$, and $\mathbb{D}^{\text{TUR}}$. Lastly, in scenario (3), we utilise the 'F&E' proportion of the GlobalWOZ dataset (Ding et al., 2022) as $\mathbb{D}^{\text{TGT}}_{mt}$. This dataset leverages a Google Translate to convert English utterances into the target language while maintaining the slot values associated with entities in English.[8]

Table 1 presents the experimental results for the three scenarios. Despite employing state-of-the-art mPLMs, we observe a substantial cross-lingual transfer gap (Hu et al., 2020) across all tasks, underscoring the significance of in-domain in-language data for the development of ToD systems. Moreover, in zero-shot setups, larger models exhibit better cross-lingual transferability for SL, ID, and DST. However, this advantage diminishes when full-sized training data becomes available.

For translate-train, we observe that the system trained using the MT-ed dataset $\mathbb{D}^{\text{TGT}}_{mt}$ underperforms the fully supervised systems $P^{\text{TGT}}(\cdot)$. Interestingly, the translate-train system performs worse than the English system $P^{\text{ENG}}(\cdot)$, even in the zero-shot cross-lingual transfer setup, for ID and SL.[9]

[8]Instead of employing an MT system to translate $\mathbb{D}^{\text{ENG}}$ into $\mathbb{D}^{\text{TGT}}_{mt}$, we utilise the GlobalWOZ dataset as our translate-train dataset. This dataset fulfils our objective by offering translated dialogues using Google Translate and provides character-level spans for slot values. We acknowledge that the quality of $\mathbb{D}^{\text{TGT}}_{mt}$ are naturally influenced by the choice of the MT system.
[9]The poor performance of SL models trained on GlobalWOZ can be attributed to the presence of erroneous span

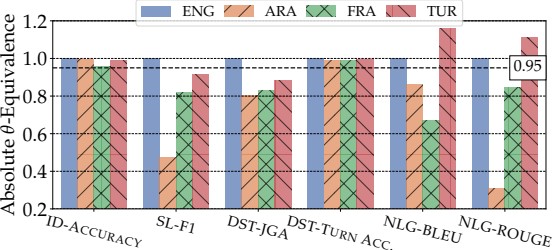

Figure 1: A cross-language comparison for absolute equivalence in performance for each supervised system $P^{\text{TGT}}(\cdot)$ with respect to the fully supervised English system $P^{\text{ENG}}(\cdot)$. The ID and SL models are based on XLM-R$_{\text{large}}$, while the DST and NLG models are based on mT5$_{\text{small}}$. The performance is reported based on task-metric pairs, such as evaluating ID with Accuracy. The 0.95-equivalence line requires a system to achieve 95% of the $P^{\text{ENG}}(\cdot)$ performance (see §3.2).

*In sum, our empirical results highlight the crucial role of acquiring high-quality in-domain in-language training data when developing ToD systems for a new language.*

**(RQ2) Intrinsic Bias in mPLMs.** In RQ2, we quantitatively evaluate the observed *performance disparity* across languages by measuring the absolute $\theta$-equivalence. Figure 1 illustrates the performance ratio of each system in relation to the English systems. This figure also highlights the absolute 0.95-equivalence threshold, which signifies the point at which a system achieves 95% of the

annotations within the GlobalWOZ dataset, demonstrating the error-prone nature of MT-based multilingual ToD generation.

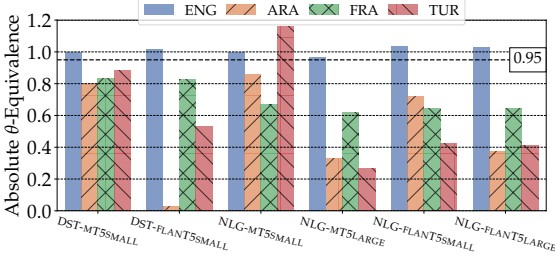

Figure 2: Absolute equivalence in performance of different mPLMs for DST (JGA) and NLG (BLEU).

performance of the fully supervised English system. For instance, the English ID system achieves an accuracy of 93.2. Thus, a target system must attain at least 88.5 accuracy to achieve 0.95-equivalence.

It is observed that the ID models in the target languages meet the 0.95-equivalence performance threshold. However, for the more complex tasks of SL, DST, and NLG, the models generally fall short of reaching the threshold, when evaluating the models using the widely accepted metrics (e.g., JGA). An exception is the Turkish NLG model, which surpasses the performance of the English model by achieving a higher BLEU score. It is important to note that the BLEU score is sensitive to the specific linguistic properties of the language being evaluated and may vary depending on the language under consideration.

In addition, DST models for all languages achieve the 0.95-equivalence performance threshold when the metric $M(\cdot)$ is Turn Accuracy. However, when evaluating the models using JGA, the target languages fail to reach the threshold. Additionally, we note that the Arabic NLG model outperforms the French one in terms of BLEU score, but falls behind in terms of ROUGE score. These findings highlight the dependence of the equivalent performance on the choice of evaluation metrics and emphasise the importance of aligning evaluation metrics with users' judgement of the system.

The attainment of absolute equivalence in performance across languages can also differ when employing different mPLMs. Figure 2 indicates that each mPLM can have varying impacts on downstream applications. Moreover, it is worth noting that the performance of mPLMs still lags behind their monolingual counterparts. For instance, by substituting the multilingual model with the monolingual Arabic-BERT_large model (Safaya et al., 2020), we achieved a F1 score of 54.9 (↑ 9.7) for

Arabic SL.

Arabic SL serves as the bottleneck across the benchmark, with a significant decrease in performance compared to other language. In the MULTI3WOZ dataset, the slot-value spans are annotated at the character level, and an exact match is required for a span to be considered correctly identified. On the other hand, Rust et al. (2021) observed that the tokenisers used in multilingual models may lead to sub-optimal performance for downstream tasks. To delve further into the analysis, we aligned the slot boundaries with the token boundaries by defining the slot span as the minimal token span that covers the entire slot in the utterance. Using this approach, the identical model achieved an improved F1 score of 81.1 (↑35.9) for Arabic SL, confirming that the sub-optimal tokenisation of XLM-R was the primary factor contributing to the performance degradation.

*In sum, despite having access to the same models and* TOD *data,* TOD *systems in target languages have not yet reached performance levels on par with English and have failed our goal of absolute 0.95-equivalence in performance. This limitation can primarily be attributed to the intrinsic bias inherent in state-of-the-art mPLMs.*

**(*RQ3*) Adaptive Bias in Few-shot Learning.** Subsequently, we investigate the impact of the scarcity of in-domain, in-language training data on *performance disparities*, recognising it as a primary source of aforementioned *adaptive bias*. Here, our objective is to collect a small-sized dataset $\mathbb{D}_{few}^{TGT}$ in the target language that enables the trained system to achieve relative 0.95-equivalence in performance ($M(P_{few}^{TGT}(\cdot)) \geq 0.95 \cdot M(P^{TGT}(\cdot))$; see §3.2).

Figure 3(a) shows the relative equivalence, that is, the performance ratio for a system trained on increasing percentage of Arabic-only data (without cross-lingual transfer) compared to fully supervised systems $P^{ARA}(\cdot)$ trained on the complete $\mathbb{D}^{ARA}$. The subsets of Arabic-only data are randomly sampled from the full Arabic training dataset. As anticipated, we find that simpler tasks like ID and SL reach relative equivalence with less than 20% of the training dataset. Beyond this point, increasing the training data size leads to only marginal performance gains. More complex tasks such as DST and NLG require more data, with relative equivalence being achieved at approximately 50% and 80% of the training data, respectively. Similar trends are observed for other languages, including English,

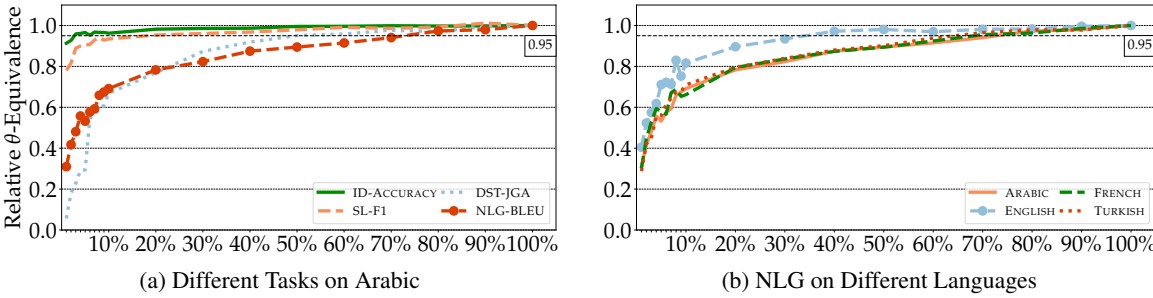

(a) Different Tasks on Arabic    (b) NLG on Different Languages

Figure 3: Relative equivalence in performance for **(a)** ID, SL, DST, and NLG for system trained on increasing percentage of randomly sampled Arabic-only training data with respect to systems $P^{\mathrm{ARA}}(\cdot)$ trained with full data and **(b)** NLG for systems trained on increasing percentage of randomly sampled monolingual data, in relation to systems trained on the full dataset. The ID and SL models are based on XLM-R$_{\mathrm{large}}$ and evaluated using Accuracy and F1 scores. The DST and NLG models are based on mT5$_{\mathrm{small}}$, and evaluated on JGA and BLEU scores.

as shown in Figure 4 (appendix). Interestingly, the English systems surpass the equivalence threshold faster compared to other target languages.

Figure 3(b) shows the relative equivalence in performance of NLG for systems trained on progressively larger proportions of monolingual training data across different languages. Notably, even when employing the same mPLM, English outperforms other target languages by achieving the 0.95-equivalence threshold with 40% of the training data. On the other hand, the remaining target languages require more than 80% of the training data to reach the threshold. Similar trends are observed for SL and DST, as illustrated in Figure 5, with the exception of ID models which all achieve the threshold with less than 5% of the training data.

Furthermore, we note the following auxiliary findings regarding the introduction of cross-lingual transfer training: **1)** We observe that cross-lingual transfer learning results in obvious gains when the amount of target language training data is limited (less than 30%). However, when evaluating systems based on the relative 0.95-equivalence threshold, this transfer learning strategy only leads to marginal or negligible improvements. **2)** The mixed cross-lingual training strategy outperforms its sequential training counterpart, which is consistent with prior research (Schmidt et al., 2022).

*In sum, the proportion of data required to achieve a relative 0.95-equivalence in performance varies depending on the complexity of the task. Simpler tasks can reach this threshold with less than 20% of the in-language data. However, more complex tasks such as DST and NLG often require a larger proportion, typically exceeding 50%.*

**(RQ4) Cost-efficiency in Multilingual ToD Col-**

**lection.** This research question considers the real life challenge faced by developers who need to create ToD systems in a new language without any in-domain, in-language data available. The objective is to maximise system performance across multiple metrics while minimising annotation cost by annotating the minimum amount of data. Specifically, we assume the availability of a large-scale source dataset $\mathbb{D}^{\mathrm{SRC}}$. Our objective is to effectively leverage $\mathbb{D}^{\mathrm{SRC}}$ to create a target language dataset $\mathbb{D}^{\mathrm{TGT}}_{few}$ that covers a subset of $\mathbb{D}^{\mathrm{TGT}}$, aiming to train a system that outperforms other possible subset selections of the same size.

For simplicity, we assume that the annotation costs for dialogues in all languages are identical. We compare different strategies for creating $\mathbb{D}^{\mathrm{SRC}}_{few}$ from $\mathbb{D}^{\mathrm{SRC}}$ and use it to create a multi-parallel target dataset $\mathbb{D}^{\mathrm{TGT}}_{few}$, thus simulating the described scenario. Specifically, we evaluate following strategies. (i) **Random Sampling:** This is a baseline strategy, where we randomly sample a subset of $\mathbb{D}^{\mathrm{SRC}}_{few}$ and create the corresponding $\mathbb{D}^{\mathrm{TGT}}_{few}$ in the target language. This strategy has been consistently applied throughout our previous analysis. (ii) **Max N-gram:** This strategy is to heuristically select dialogues that maximise the trigram diversity of $\mathbb{D}^{\mathrm{SRC}}_{few}$. The objective is to enhance the diversity and coverage of language patterns in $\mathbb{D}^{\mathrm{TGT}}_{few}$. (iii) **Equal Domain:** In this strategy, we sample an equivalent number of dialogues for each domain. The goal is to ensure a balanced representation of all domains in $\mathbb{D}^{\mathrm{TGT}}_{few}$. (iv) **Equal Slot:** This strategy focuses on sampling a set of dialogues while maintaining a balanced distribution of slot occurrences. (v) **Max Length:** This strategy involves selecting a list of dialogues with the longest length, i.e., the highest

| Strategy | ID | SL | DST | NLG |
|---|---|---|---|---|
| | Accuracy | F1 | JGA | BLEU |
| Random Sampling | 87.9 | 65.2 | 20.7 | 10.4 |
| Max N-gram | **88.8** | **66.5** | 23.6 | **12.2** |
| Equal Domain | 87.2 | 65.3 | 21.1 | 10.1 |
| Equal Slot | 87.9 | 65.0 | 26.2 | 11.3 |
| Max Length | 88.3 | 66.4 | **26.7** | 11.5 |

Table 2: The average performance of the models for each task across three target languages: Arabic, French, and Turkish. Each model was trained using 5% of the target language data, which was generated using each data creation strategy introduced in *RQ4*. The summarised results presented in this table are derived from Table 10, Table 12, and Table 13 in the Appendix. The ID and SL models are built upon XLM-R$_{large}$, while the DST and NLG models are based on mT5$_{small}$.

number of utterances.

Table 2 presents the averaged task performance for the three target languages, considering a few-shot setup where each model is trained on 5% of the full dataset $\mathbb{D}^{TGT}$. The results indicate that the Max N-gram and Max Length strategies consistently outperform the baseline Random Sampling strategy across all tasks. Notably, the Max N-gram strategy achieves the highest performance for ID, SL, and NLG tasks. These findings emphasise the potential of strategic annotation budget allocation in multilingual dialogue collection, which can lead to enhanced model performance without incurring additional annotation costs.

Furthermore, we present the following auxiliary findings. **1)** We found that there is a negligible performance gain when collecting a full-sized (100%) validation set compared to a smaller (e.g., 10%) validation set. **2)** When developing ToD systems for multiple languages simultaneously, we observed that collecting distinct (mutually exclusive) sets of dialogues in each language, without overlapping dialogue patterns, led to marginal enhancements in NLU in a cross-lingual transfer setup. However, this observed pattern did not extend to NLG.

*In sum, our findings show the benefits of strategic budget allocation in multilingual dialogue collection, which can enhance model performance without additional costs. Particularly, the Max N-gram strategy, which involves creating a target dataset $\mathbb{D}^{TGT}_{few}$ from a subset $\mathbb{D}^{SRC}_{few}$ that maximises both trigram and lexical coverage, leads to improved performance compared to the random baseline.*

## 6 Conclusion and Future Work

We presented a systematic analysis of p*erformance disparities* in modern multilingual ToD systems, utilising proposed quantitative measures of *absolute* and *relative equivalence* in performance. By addressing four key research questions, our empirical investigation revealed the presence of *adaptation* and *intrinsic biases* in current ToD systems and provided insights on how to best carry out data collection and system development for new languages. This study opens up new avenues for future research, including mitigating biases, improving system robustness, and expanding ToD systems across languages and domains.

## Limitations

The limitations of this work are primarily related to the assumptions made during our analysis. Firstly, in §3.1, we rely on established automatic evaluation metrics and assume a strong correlation between these metrics and human judgement of system performance. While this assumption has been widely adopted in previous studies (Blasi et al., 2022; Khanuja et al., 2023, *inter alia*), it is important to acknowledge the ongoing challenge in dialogue system evaluation: Recent research (Yeh et al., 2021; Mehri et al., 2022) has highlighted that automatic evaluation metrics demonstrate only moderate correlation with human judgement. This indicates the need for further development of dialogue evaluation. On the other hand, it is worth noting that our proposed notions of $\theta$-equivalence are not restricted to any specific metric but are generally applicable, including to further metrics.

Another limitation is the potential misinterpretation of $\theta$-equivalence as implying that $P^{TGT}(\cdot)$ is as $\theta$-good as $P^{SRC}(\cdot)$. It is important to note that achieving $\theta$-equivalence does not indicate that the system in the target language, $P^{TGT}(\cdot)$, possesses '$\theta$-proportion' of capability compared to the system in the source language, $P^{SRC}(\cdot)$. The concept of equivalence in performance focuses on providing an indicator of the *performance disparities* across languages or within a target language, as well as aims to minimise these disparities. Similar to evaluation metrics, as an example, a generation system achieving a 20 BLEU score does not necessarily imply that it is twice as good as a system obtaining a 10 BLEU score. Our proposed measures should be interpreted cautiously, considering the nuanced nature of dialogue evaluation and the potential non-

linear relationship between metric scores and user experience (Blasi et al., 2022).

Besides, in §5 *(RQ4)*, we abstract away many fine-grained details in a real-word data collection project. For example, we assume that annotating a dialogue for all languages incurs uniform costs. While this assumption simplifies the analysis, it may not accurately reflect the real-world scenario. The actual cost of annotating dialogues can vary across languages due to factors such as the availability of native or bilingual annotators. Furthermore, some strategies tested in our study, such as the Max Length strategy, may result in increased data collection time and subsequently higher costs. We hope that our findings could inspire future work to delve into more sophisticated annotation strategies and conduct a more fine-grained analysis, considering these finer details of real-world data collection. Given the scope of this paper, we were unable to delve into the intricacies of cost variations in multilingual dialogue annotation.

The scope of languages analysed in this study is limited by the availability of training data. As mentioned earlier, the MULTI3WOZ dataset provides us with the opportunity to conduct this type of study for the first time. Therefore, our analysis is based on this single dataset, which includes four languages: Arabic, English, French, and Turkish. While these languages are representative of different language families and feature a diverse range of linguistic properties, we recognise that our findings are derived from a small sample of world languages. Future research might (and should) contribute more resources, particularly for under-represented languages, and expand the analysis to a broader linguistic landscape. Furthermore, a fully inclusive dialogue system should consider not only text input but also other modalities, such as spoken and sign languages. We also acknowledge that the limitations of training data have restricted the analysis to text input only, thus discarding potential disparities stemming from other relevant tools such as automatic speech recognition (Babu et al., 2022).

Finally, we acknowledge that our experimental results are derived from a single run, rather than multiple runs with varying random seeds. In this study, we have conducted a comprehensive set of experiments including four different languages and spanning across a range of tasks. These tasks include simpler tasks such as ID and SL, as well as more complex tasks like DST and NLG. Conse-

quently, the cumulative computational budget of these experiments exceeded 5,000 GPU hours on state-of-the-art GPUs. Considering the importance of conserving valuable computing resources and minimising energy consumption, we did not to run parallel experiments with different random seeds. However, we would like to emphasise that, despite based on a single run, all our main claims remain consistent across all languages and tasks, thereby affirming their reliability and generalisability.

## Ethics Statement

The experimental study obtained full Ethics Approval from the University of Cambridge in advance. No human participants or personal data were directly involved in this study. However, our models leverage two data sources: the MULTI3WOZ dataset and the pre-training data of each mPLM employed in this study. Hu et al. (2023) highlights that the creation and publication of MULTI3WOZ comply with the EU General Data Protection Regulation (GDPR). Particularly, this dataset consists solely of hypothetical dialogues in which the domains and content have been restricted and predefined, minimising the risk of personal data being present. On the other hand, it is important to acknowledge that although these PLMs are publicly available, there exists a potential risk of privacy violations (Brown et al., 2022; Carlini et al., 2021).

## Acknowledgements

Songbo Hu is supported by the Cambridge International Scholarship. Han Zhou is supported by the UK Research and Innovation (UKRI) Frontier Research Grant EP/Y031350/1 (the UK government's funding guarantee for ERC Advanced Grants) at the University of Cambridge. Anna Korhonen acknowledges the support of the UK EPSRC grant EP/T02450X/1 and the UKRI Frontier grant EP/Y031350/1. Ivan Vulić acknowledges the support of a personal Royal Society University Research Fellowship *'Inclusive and Sustainable Language Technology for a Truly Multilingual World'* (no 221137; 2022–). The team recognises support from Huawei Technologies.

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

## A  Experimental Details

**Experiments on ID.** Our ID model is implemented as a multi-class classifier. At each dialogue turn $t$, an mPLM model encodes the concatenation of the previous two historical utterances ($\mathbf{u}_{t-2}$ and $\mathbf{u}_{t-1}$) along with the current utterance ($\mathbf{u}_t$) to provide embedding vectors at the sequence. For each domain-intent pair $d$-$i$ defined by the ontology, the representation of the '' token is subsequently projected down to two logits and passed through a softmax layer to form a Bernoulli distribution indicating if $d$-$i$ appears in the $\mathbf{u}_t$. Unless otherwise specified, all of our models, including ID models, in this paper are implemented using the HuggingFace repository (Wolf et al., 2020). All ID experiments were run on a single A100 80 GiB GPU and a 32-core vCPU.

**Table 3** presents the selected hyper-parameters for the conducted experimental study, including the ID models.

**Table 4** lists all the PLMs we used in this work, along with their respective checkpoints in the Huggingface repository.

**Table 5** shows the time consumption of the models for all tasks in the experimental study, including the ID models, The time consumption is measured based on five independent runs on the MULTI3WOZ dataset.

**SL.** Our SL model is implemented as a sequence tagger. Precisely, at each dialog turn $t$, the model encodes the concatenation of the previous two utterances ($\mathbf{u}_{t-2}$ and $\mathbf{u}_{t-1}$) along with the current utterance ($\mathbf{u}_t$) to provide embedding vectors at token levels. We adopt the widely-used BIO labelling scheme to annotate each token in the user's utterance. Specifically, each token is labelled with either B-$d$-$i$-$s$ (e.g., *B-Restaurant-Inform-Food*), denoting the beginning of a slot-value pair with the corresponding slot name, I-$d$-$i$-$s$ indicating it is inside the slot-value, or O indicating that the token is not associated with any slot-value pair. The prediction is performed through token classification.[10] We implemented our SL model using the HuggingFace repository and conducted the experiments on a single A100 80 GiB GPU and a 32-core vCPU. The selected hyper-parameter, mPLMs used, and the time consumption of our SL experiments are

---

[10]https://huggingface.co/tasks/token-classification

| Hyper-parameter | Value |
|---|---|
| **ID: XLM-R$_{base}$, XLM-R$_{large}$** | |
| batch size | 128 |
| learning rate | 2e-5 |
| weight decay | 0.1 |
| evaluation per steps | 500 |
| max training steps | 20000 |
| max context char length | 150 |
| early stopping patience | 2 |
| **SL: XLM-R$_{base}$, XLM-R$_{large}$** | |
| batch size | 128 |
| learning rate | 2e-5 |
| weight decay | 0.1 |
| evaluation per steps | 500 |
| max training steps | 20000 |
| max context char length | 150 |
| early stopping patience | 2 |
| **DST: mT5$_{small}$, Flan-T5$_{small}$** | |
| batch size | 8 |
| learning rate | 1e-5 |
| max training epochs | 5 |
| early stopping patience | 2 |
| **DST: mT5$_{large}$, Flan-T5$_{large}$** | |
| batch size | 4 |
| learning rate | 1e-5 |
| number of training epochs | 1 |
| **NLG: mT5$_{small}$, Flan-T5$_{small}$** | |
| batch size | 32 |
| learning rate | 1e-3 |
| weight decay | 0.01 |
| evaluation per steps | 2000 |
| max training steps | 40000 |
| max context char length | 200 |
| early stopping patience | 2 |
| **NLG: mT5$_{large}$, Flan-T5$_{large}$** | |
| batch size | 16 |
| learning rate | 1e-3 |
| weight decay | 0.01 |
| evaluation per steps | 2000 |
| max training steps | 40000 |
| max context char length | 200 |
| early stopping patience | 2 |

Table 3: Model hyper-parameters for each task-model pair. For example, ID: XLM-R$_{base}$, XLM-R$_{large}$ section in this table shows the hyper-parameters for ID models back-boned with XLM-R$_{base}$ and XLM-R$_{large}$. To select the optimal model checkpoint, we employ early stopping and select the one with the best validation performance, measured by F1, F1, validation loss, and BLEU for ID, SL, DST, and NLG, respectively. Unless explicitly specified, all other hyper-parameters are set to their default values as defined in the HuggingFace Transformers or T5DST repository.

| Model | HuggingFace Checkpoint |
|---|---|
| XLM-R$_{base}$ | xlm-roberta-base |
| XLM-R$_{large}$ | xlm-roberta-large |
| mT5$_{small}$ | google/mt5-small |
| mT5$_{large}$ | google/mt5-large |
| Flan-T5$_{small}$ | google/flan-t5-small |
| Flan-T5$_{large}$ | google/flan-t5-large |
| Arabic-BERT$_{large}$ | asafaya/bert-large-arabic |

Table 4: The employed mPLMs in our experimental study and their Huggingface Checkpoints.

| Setup | Time Consumption |
|---|---|
| **ID: XLM-R$_{base}$** | |
| Training per 500 steps | 3:26 |
| Inference on *full test* | 0:06 |
| **ID: XLM-R$_{large}$** | |
| Training per 500 steps | 10:29 |
| Inference on *full test* | 0:20 |
| **SL: XLM-R$_{base}$** | |
| Training per 500 steps | 3:21 |
| Inference on *full test* | 0:45 |
| **SL: XLM-R$_{large}$** | |
| Training per 500 steps | 10:36 |
| Inference on *full test* | 1:24 |
| **DST: mT5$_{small}$** | |
| Training per epoch | 3:56 |
| Inference on *full test* | 6:53 |
| **DST: mT5$_{large}$** | |
| Training per epoch* | 30:54:23 |
| Inference on *full test*\* | 17:17 |
| **NLG: mT5$_{small}$** | |
| Training per 2000 steps | 7:12 |
| Inference on *full test* | 0:56 |
| **NLG: mT5$_{large}$** | |
| Training per 2000 steps | 37:02 |
| Inference on *full test* | 3:22 |

Table 5: Time consumption of models for each dialogue task. It was computed as an average of 5 runs on a machine with a single A100 80 GiB GPU and a 32-core vCPU. ($*$) For the DST experiments utilising mT5$_{large}$ models, we conducted the experiments on a machine equipped with two A100 80 GiB GPUs. We report this run time based on a single run to account for its significant time consumption.

summarised in Table 3, Table 4, and Table 5, respectively.

**DST.** For detailed information about the DST model design and training methodology, we refer readers to the original work by Lin et al. (2021a). Our implementation is based on the official T5DST

GitHub Repository.[11] For the DST experiments utilising $mT5_{small}$ models, we run them on a single A100 80 GiB GPU and a 32-core vCPU. Due to the lack of support in the official code for evaluating and saving models per training steps, we adopt a specific training protocol. In the fully supervised setup, we train DST models for a fixed number of 5 epochs. On the other hand, in the few-shot setup, we train the models for approximately the same number of epochs that corresponds to a similar number of training steps as 5 epochs. For instance, if we train models with 10% of the full training data, we train them for a maximum of 50 epochs. For the DST experiments utilising $mT5_{large}$ models, we conducted the experiments on a machine equipped with two A100 80 GiB GPUs. However, due to the significant time consumption, the models were only trained for 1 epoch. We report the selected hyper-parameter, mPLMs used, and the time consumption of our DST experiments in Table 3, Table 4, and Table 5, respectively.

**NLG.** The NLG model is implemented using the HuggingFace repository. All NLG experiments were run on a single A100 80 GiB GPU and a 32-core vCPU. We report the selected hyper-parameter, mPLMs used, and the time consumption of our DST experiments in Table 3, Table 4, and Table 5, respectively.

## B  Additional Results on MULTI3WOZ

To solidify the empirical findings of our paper, we present supplementary experimental results that provide additional support for our main claims, as well as our auxiliary findings discussed in §5.

### (*RQ3*) Adaptive Bias in Few-shot Learning.

**Figure 4** visually presents the amount of in-language training data necessary to attain relative 0.95-equivalence in performance for four tasks: ID, SL, DST, and NLG across three languages: English, French, and Turkish. These results validate our claim stated in §5, as also supported by Figure 3(a). They demonstrate that our claim extends beyond the Arabic language and are applicable to other languages as well.

**Figure 5** shows the amount of in-language training data necessary to attain relative 0.95-equivalence in performance for each languages across three tasks: ID, SL, and DST. These results validate our claim

stated in §5, as also supported by Figure 3(b). They demonstrate that our claim extends beyond a single NLG task and are applicable to other tasks as well.

**Figure 6** demonstrates the benefits of incorporating cross-lingual transfer training in dialogue tasks. We observe that the cross-lingual transfer learning from $\mathbb{D}^{ENG}$ yields substantial improvements when the amount of target language training data is limited (less than 30%). However, when assessing systems based on the relative 0.95-equivalence threshold, this transfer learning strategy only results in marginal or negligible improvements. While support our supporting for our auxiliary finding **1)**, these results once again emphasise the significance of in-domain, in-language training data. They also present an open research challenge to explore the optimal utilisation of cross-lingual transfer learning in conjunction with a larger quantity of in-language training data.

**Figure 7** compares the benefits of incorporating two types of cross-lingual transfer training strategies in dialogue tasks. In the figure, the term 10% of training data denotes the mixed training strategy using the complete $\mathbb{D}^{ENG}$ dataset combined with 10% of the $\mathbb{D}^{ARA}$ dataset in the same training epoch. This mixed training strategy outperforms its sequential training counterpart, where the model is initialised based on $\mathbb{P}^{ENG}$, for ID and NLG tasks. In addition, **Figure 8** confirms that our findings holds across languages. These findings align with previous results (Schmidt et al., 2022). These results provide support for our auxiliary finding **2)**.

**Table 6**, **Table 7**, **Table 8**, and **Table 9** present the performance of ID, SL, DST, and NLG tasks across different languages: Arabic, English, French, and Turkish, respectively. These tables specifically demonstrate the model's performance in a few-shot setup, where the proportion of available training data increases progressively. The discussion and analyses regarding the research question are derived from the performance presented in these tables.

### (*RQ4*) Cost-efficiency in Multilingual TOD Collection.

**Figure 9** illustrates the comparative performance of SL models trained on different proportions of data in Arabic, French, and Turkish, using data created with two strategies: Random Sampling and Max N-gram.

---

[11] https://github.com/facebookresearch/Zero-Shot-DST

| % of Training Data | ID | | SL | | | DST | | | NLG | | |
|---|---|---|---|---|---|---|---|---|---|---|---|
| | Accuracy | F1 | Precision | Recall | F1 | JGA | Turn Acc. | F1 | BLEU | ROUGE | METEOR |
| Arabic | | | | | | | | | | | |
| 1% | 84.7 | 89.8 | 33.4 | 37.5 | 35.3 | 2.7 | 84.9 | 46.9 | 5.5 | 3.6 | 16.5 |
| 2% | 85.9 | 90.6 | 34.0 | 40.4 | 36.9 | 8.7 | 89.3 | 62.7 | 7.5 | 3.9 | 20.4 |
| 3% | 88.9 | 92.4 | 38.0 | 42.8 | 40.2 | 11.0 | 90.8 | 67.0 | 8.6 | 7.8 | 22.1 |
| 4% | 89.1 | 92.7 | 38.1 | 43.7 | 40.7 | 13.9 | 91.7 | 69.9 | 9.9 | 6.2 | 24.4 |
| 5% | 89.5 | 92.8 | 38.0 | 44.2 | 40.9 | 14.1 | 91.9 | 70.3 | 9.5 | 5.9 | 23.8 |
| 6% | 88.3 | 92.0 | 38.1 | 44.5 | 41.0 | 26.1 | 94.3 | 78.9 | 10.3 | 7.9 | 25.2 |
| 7% | 89.8 | 93.0 | 39.2 | 45.2 | 42.0 | 27.2 | 94.4 | 80.2 | 10.6 | 7.4 | 25.4 |
| 8% | 89.6 | 93.0 | 40.5 | 44.8 | 42.5 | 28.3 | 94.6 | 80.5 | 11.8 | 9.2 | 27.4 |
| 9% | 89.6 | 92.7 | 39.2 | 45.2 | 42.0 | 29.2 | 94.9 | 81.3 | 12.1 | 8.1 | 27.9 |
| 10% | 89.2 | 92.8 | 39.8 | 45.0 | 42.2 | 32.0 | 95.2 | 82.5 | 12.4 | 8.4 | 28.1 |
| 20% | 91.0 | 93.9 | 40.7 | 45.9 | 43.1 | 36.9 | 95.8 | 95.8 | 14.0 | 10.3 | 30.3 |
| 30% | 91.4 | 94.1 | 40.9 | 46.4 | 43.5 | 41.9 | 96.3 | 87.2 | 14.8 | 11.3 | 31.6 |
| 40% | 91.5 | 94.3 | 40.8 | 47.2 | 43.8 | 44.0 | 96.5 | 87.8 | 15.7 | 12.7 | 32.8 |
| 50% | 92.3 | 94.7 | 41.5 | 47.4 | 44.3 | 45.6 | 96.6 | 88.2 | 16.0 | 13.4 | 33.4 |
| 60% | 92.4 | 94.7 | 41.8 | 48.2 | 44.8 | 46.0 | 96.7 | 88.5 | 16.4 | 13.6 | 33.9 |
| 70% | 92.6 | 95.0 | 41.9 | 47.6 | 44.6 | 46.8 | 96.8 | 89.0 | 16.9 | 13.8 | 34.7 |
| 80% | 92.5 | 94.9 | 42.0 | 48.1 | 44.9 | 46.5 | 96.7 | 88.9 | 17.4 | 14.7 | 35.3 |
| 90% | 92.5 | 94.9 | 43.4 | 48.4 | 45.8 | 47.7 | 96.9 | 89.4 | 17.6 | 15.4 | 35.4 |
| 100% | 92.7 | 95.0 | 42.4 | 48.5 | 45.2 | 47.9 | 96.9 | 89.4 | 17.9 | 15.0 | 36.0 |

Table 6: The performance of ID, SL, DST, and NLU models trained on an increasing percentage of Arabic data. The ID and SL models are built upon XLM-R$_{large}$, and the DST and NLG models are based on mT5$_{small}$.

| % of Training Data | ID | | SL | | | DST | | | NLG | | |
|---|---|---|---|---|---|---|---|---|---|---|---|
| | Accuracy | F1 | Precision | Recall | F1 | JGA | Turn Acc. | F1 | BLEU | ROUGE | METEOR |
| English | | | | | | | | | | | |
| 1% | 81.6 | 88.3 | 86.3 | 88.3 | 87.3 | 17.0 | 91.1 | 71.5 | 8.4 | 29.7 | 26.9 |
| 2% | 85.4 | 91.2 | 89.2 | 91.6 | 90.4 | 24.5 | 93.5 | 78.0 | 10.9 | 34.7 | 31.5 |
| 3% | 85.7 | 91.1 | 89.5 | 93.2 | 91.3 | 30.5 | 94.6 | 80.9 | 12.0 | 36.3 | 33.0 |
| 4% | 88.3 | 92.8 | 91.3 | 93.9 | 92.6 | 32.1 | 95.0 | 82.5 | 12.9 | 38.2 | 34.7 |
| 5% | 88.8 | 93.4 | 91.2 | 94.0 | 92.6 | 39.8 | 96.1 | 86.1 | 14.8 | 41.2 | 37.5 |
| 6% | 90.0 | 94.0 | 92.3 | 93.5 | 92.9 | 35.8 | 95.7 | 84.8 | 15.0 | 41.9 | 37.9 |
| 7% | 90.0 | 94.0 | 91.9 | 93.9 | 92.9 | 40.4 | 96.2 | 86.7 | 14.9 | 40.9 | 36.4 |
| 8% | 89.5 | 93.9 | 91.8 | 94.7 | 93.2 | 44.3 | 96.5 | 88.2 | 17.3 | 44.1 | 39.9 |
| 9% | 88.7 | 93.5 | 92.1 | 95.3 | 93.7 | 42.2 | 96.4 | 87.8 | 15.7 | 41.7 | 38.0 |
| 10% | 91.0 | 94.9 | 92.1 | 95.1 | 93.5 | 43.2 | 96.5 | 87.5 | 17.0 | 43.7 | 39.7 |
| 20% | 91.1 | 94.9 | 93.1 | 95.2 | 94.1 | 47.0 | 96.8 | 89.5 | 18.7 | 46.2 | 42.3 |
| 30% | 91.3 | 95.0 | 93.3 | 95.1 | 94.5 | 54.2 | 97.4 | 91.8 | 19.5 | 46.5 | 42.5 |
| 40% | 92.4 | 95.7 | 93.6 | 94.8 | 94.2 | 55.2 | 97.5 | 92.1 | 20.3 | 47.4 | 43.1 |
| 50% | 92.7 | 95.8 | 94.3 | 94.9 | 94.6 | 56.4 | 97.7 | 92.8 | 20.4 | 47.3 | 43.0 |
| 60% | 92.4 | 95.6 | 93.9 | 95.7 | 94.8 | 56.6 | 97.7 | 92.9 | 20.2 | 47.4 | 43.6 |
| 70% | 92.8 | 95.9 | 94.2 | 95.8 | 95.0 | 58.4 | 97.8 | 93.2 | 20.5 | 47.6 | 43.6 |
| 80% | 93.0 | 96.0 | 94.6 | 95.9 | 95.2 | 58.7 | 97.8 | 93.5 | 20.5 | 47.7 | 43.2 |
| 90% | 92.5 | 95.7 | 95.0 | 95.5 | 95.3 | 59.3 | 97.8 | 93.6 | 20.8 | 48.1 | 43.9 |
| 100% | 93.2 | 96.1 | 94.6 | 95.7 | 95.1 | 59.8 | 97.9 | 93.5 | 20.8 | 48.4 | 44.1 |

Table 7: The performance of ID, SL, DST, and NLU models trained on an increasing percentage of English data. The ID and SL models are built upon XLM-R$_{large}$, and the DST and NLG models are based on mT5$_{small}$.

| % of | ID | | SL | | | DST | | | NLG | | |
|---|---|---|---|---|---|---|---|---|---|---|---|
| Training Data | Accuracy | F1 | Precision | Recall | F1 | JGA | Turn Acc. | F1 | BLEU | ROUGE | METEOR |
| French | | | | | | | | | | | |
| 1% | 79.5 | 86.5 | 64.4 | 64.8 | 64.6 | 3.0 | 83.9 | 46.3 | 4.2 | 21.4 | 16.9 |
| 2% | 80.6 | 87.3 | 70.1 | 71.1 | 70.6 | 9.8 | 89.8 | 63.9 | 6.3 | 27.1 | 22.2 |
| 3% | 85.8 | 90.2 | 72.4 | 71.9 | 72.1 | 14.9 | 91.9 | 70.0 | 7.3 | 29.7 | 24.6 |
| 4% | 85.9 | 90.7 | 72.0 | 72.4 | 72.2 | 23.9 | 93.8 | 78.4 | 8.3 | 31.6 | 26.2 |
| 5% | 85.7 | 90.6 | 71.9 | 72.4 | 72.1 | 20.7 | 93.3 | 76.0 | 8.1 | 30.5 | 25.7 |
| 6% | 85.8 | 90.7 | 72.4 | 74.1 | 73.3 | 25.4 | 94.3 | 79.6 | 7.9 | 30.6 | 25.3 |
| 7% | 85.3 | 90.6 | 73.4 | 75.2 | 74.3 | 30.4 | 94.8 | 81.3 | 9.4 | 33.3 | 27.8 |
| 8% | 86.5 | 91.0 | 75.0 | 74.4 | 74.7 | 27.1 | 94.5 | 79.6 | 9.6 | 34.1 | 28.7 |
| 9% | 86.0 | 90.8 | 73.3 | 74.1 | 73.7 | 33.0 | 95.3 | 83.3 | 9.1 | 33.6 | 28.1 |
| 10% | 86.3 | 90.9 | 74.0 | 72.9 | 73.5 | 31.4 | 95.1 | 82.5 | 9.2 | 33.5 | 28.0 |
| 20% | 87.2 | 91.7 | 75.5 | 77.3 | 76.4 | 38.2 | 95.9 | 85.9 | 11.1 | 36.9 | 31.3 |
| 30% | 88.2 | 92.2 | 76.3 | 78.0 | 77.1 | 43.3 | 96.4 | 87.8 | 11.6 | 37.8 | 32.2 |
| 40% | 89.0 | 92.8 | 77.1 | 77.0 | 77.1 | 44.7 | 96.6 | 88.4 | 12.2 | 37.6 | 32.2 |
| 50% | 89.0 | 92.8 | 78.1 | 78.3 | 78.2 | 44.4 | 96.6 | 88.5 | 12.5 | 38.5 | 33.0 |
| 60% | 88.5 | 92.6 | 76.3 | 78.6 | 77.4 | 46.4 | 96.7 | 88.9 | 12.9 | 39.1 | 33.4 |
| 70% | 89.0 | 92.9 | 78.6 | 78.3 | 78.5 | 47.8 | 96.9 | 89.5 | 13.3 | 39.5 | 34.0 |
| 80% | 89.2 | 93.0 | 77.8 | 78.8 | 78.3 | 46.4 | 96.8 | 89.3 | 13.4 | 39.9 | 34.4 |
| 90% | 89.2 | 92.9 | 78.4 | 78.4 | 78.4 | 48.8 | 97.0 | 89.9 | 13.7 | 40.4 | 34.9 |
| 100% | 89.2 | 93.0 | 76.9 | 79.2 | 78.0 | 49.7 | 97.0 | 90.1 | 13.9 | 40.9 | 35.2 |

Table 8: The performance of ID, SL, DST, and NLU models trained on an increasing percentage of French data. The ID and SL models are built upon XLM-R$_{large}$, and the DST and NLG models are based on mT5$_{small}$.

| % of | ID | | SL | | | DST | | | NLG | | |
|---|---|---|---|---|---|---|---|---|---|---|---|
| Training Data | Accuracy | F1 | Precision | Recall | F1 | JGA | Turn Acc. | F1 | BLEU | ROUGE | METEOR |
| Turkish | | | | | | | | | | | |
| 1% | 79.9 | 86.8 | 74.8 | 76.5 | 75.7 | 3.6 | 85.7 | 48.1 | 6.9 | 27.1 | 24.5 |
| 2% | 84.9 | 90.2 | 81.1 | 82.0 | 81.6 | 11.7 | 90.4 | 68.4 | 10.2 | 32.9 | 29.6 |
| 3% | 86.8 | 91.1 | 81.0 | 82.2 | 81.6 | 20.2 | 92.8 | 76.2 | 10.8 | 34.8 | 2 30.9 |
| 4% | 87.4 | 91.7 | 82.5 | 83.7 | 83.1 | 27.3 | 94.5 | 81.0 | 13.5 | 38.5 | 34.4 |
| 5% | 88.4 | 92.4 | 81.8 | 83.4 | 82.6 | 27.3 | 94.4 | 81.2 | 13.6 | 39.1 | 34.6 |
| 6% | 88.7 | 92.9 | 82.9 | 83.7 | 83.3 | 32.9 | 95.3 | 83.5 | 14.9 | 40.9 | 36.7 |
| 7% | 88.1 | 92.1 | 82.9 | 84.2 | 83.5 | 33.1 | 95.2 | 83.6 | 15.6 | 42.1 | 37.6 |
| 8% | 88.7 | 92.7 | 83.7 | 85.0 | 84.3 | 32.1 | 95.2 | 84.1 | 16.3 | 43.0 | 38.6 |
| 9% | 88.7 | 92.7 | 83.9 | 85.0 | 84.4 | 34.8 | 95.5 | 84.6 | 16.4 | 43.8 | 39.1 |
| 10% | 88.3 | 92.2 | 84.6 | 84.9 | 84.7 | 35.9 | 95.6 | 85.1 | 17.2 | 44.6 | 39.9 |
| 20% | 89.6 | 93.2 | 84.5 | 85.8 | 85.1 | 44.1 | 96.5 | 87.9 | 19.3 | 47.4 | 42.5 |
| 30% | 90.7 | 94.1 | 84.3 | 84.6 | 84.4 | 46.3 | 96.7 | 89.0 | 20.3 | 48.9 | 43.9 |
| 40% | 91.5 | 94.5 | 85.2 | 86.0 | 85.6 | 48.9 | 96.8 | 89.6 | 21.3 | 49.9 | 44.9 |
| 50% | 91.4 | 94.5 | 87.1 | 85.6 | 86.4 | 48.2 | 96.9 | 89.7 | 21.8 | 50.6 | 45.7 |
| 60% | 91.8 | 94.7 | 86.5 | 86.3 | 86.4 | 50.1 | 97.1 | 90.1 | 22.7 | 51.9 | 47.0 |
| 70% | 91.7 | 94.6 | 85.7 | 87.1 | 86.4 | 50.8 | 97.1 | 90.5 | 23.4 | 52.5 | 47.4 |
| 80% | 91.6 | 94.7 | 86.8 | 87.1 | 87.0 | 51.3 | 97.1 | 90.5 | 23.7 | 53.2 | 48,1 |
| 90% | 92.0 | 94.9 | 86.6 | 86.8 | 86.7 | 52.1 | 98.1 | 90.5 | 23.7 | 53.2 | 48.2 |
| 100% | 92.2 | 95.0 | 86.6 | 87.6 | 87.1 | 52.9 | 97.3 | 91.2 | 24.2 | 53.7 | 48.6 |

Table 9: The performance of ID, SL, DST, and NLU models trained on an increasing percentage of Turkish data. The ID and SL models are built upon XLM-R$_{large}$, and the DST and NLG models are based on mT5$_{small}$.

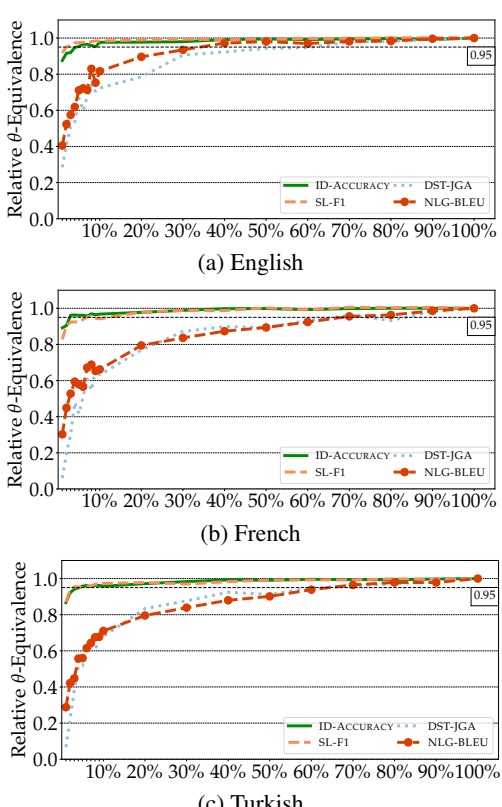

Figure 4: Relative equivalence in performance for ID, SL, DST, and NLG for system trained on increasing percentage of **(a)** English-only, **(b)** French-only, and **(c)** Turkish-only training data with respect to fully supervised systems, namely $P^{\mathrm{ENG}}(\cdot)$, $P^{\mathrm{FRA}}(\cdot)$, and $P^{\mathrm{TUR}}(\cdot)$, trained on full sized monolingual in-language data. The ID and SL models are based on $\mathrm{XLM{-}R_{large}}$, while the DST and NLG models are based on $\mathrm{mT5_{small}}$.

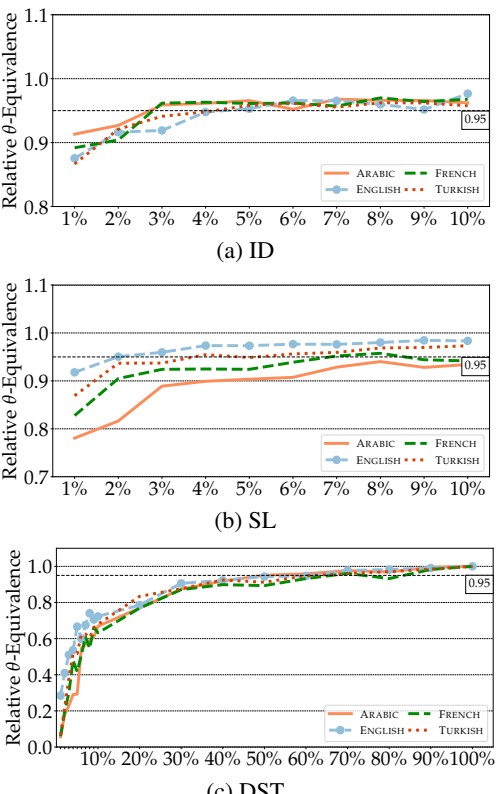

Figure 5: The relative equivalence in performance of different tasks: **(a)** ID, **(b)** SL, and **(c)** DST, is evaluated for systems trained on an increasing percentage of monolingual training data, in relation to fully supervised systems trained on the full monolingual dataset. The ID and SL models are based on $\mathrm{XLM{-}R_{large}}$, while the NLG models are based on $\mathrm{mT5_{small}}$.

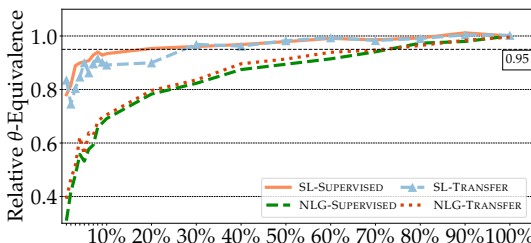

Figure 6: The comparative performance of Arabic SL and NLG systems is evaluated by training them on increasing percentage of Arabic-only training data, with (transfer) and without (supervised) cross-lingual transfer training from $\mathbb{D}^{\mathrm{ENG}}$. The evaluation is conducted with respect to fully supervised systems $P^{\mathrm{ARA}}(\cdot)$ trained on the complete $\mathbb{D}^{\mathrm{ARA}}$ dataset.

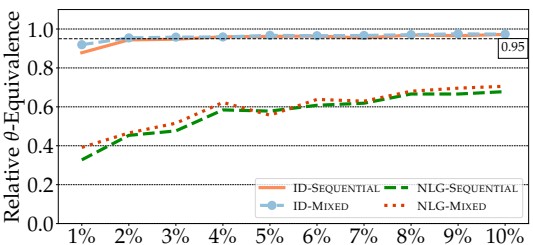

Figure 7: The performance of Arabic SL and NLG systems is compared by training them on increasing percentages of Arabic-only training data and full $\mathbb{D}^{\text{ENG}}$. Two cross-lingual transfer training strategies, namely sequential and mixed, are employed and compared. The evaluation is conducted by comparing the systems' performance to that of fully supervised systems $P^{\text{ARA}}(\cdot)$ trained on the complete Arabic training dataset $\mathbb{D}^{\text{ARA}}$. ID is evaluated using Accuracy, while NLG is evaluated using BLEU. The ID model is based on XLM-R$_{\text{base}}$, and the NLG model is based on mT5$_{\text{small}}$.

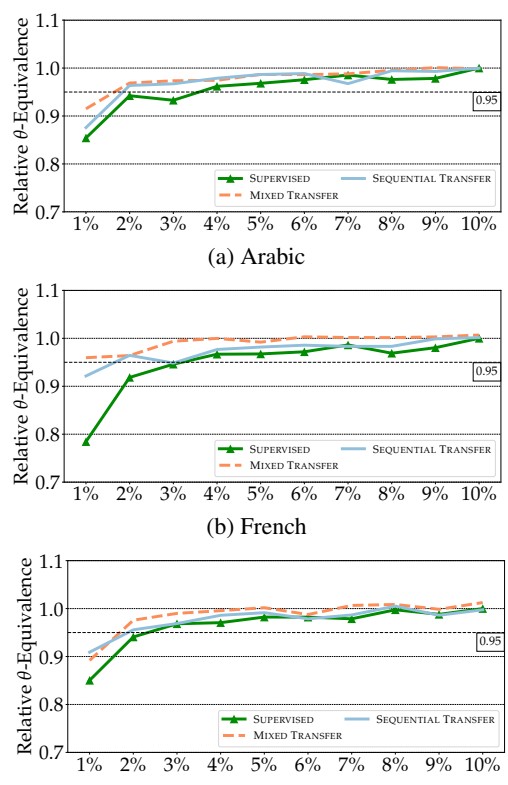

(a) Arabic

(b) French

(c) Turkish

Figure 8: Relative equivalence in performance for ID for system trained on increasing percentage of in-language data for **(a)** Arabic, **(b)** French, and **(c)** Turkish, with respect to fully supervised systems, namely $P^{\text{ENG}}(\cdot)$, $P^{\text{FRA}}(\cdot)$, and $P^{\text{TUR}}(\cdot)$, trained on full sized monolingual in-language data. Two cross-lingual transfer training strategies, namely sequential and mixed, are employed and compared. The ID models are based on XLM-R$_{\text{base}}$ and evaluated with Accuracy.

**Figure 10** illustrates the comparative performance of SL and DST models trained on different proportions of Arabic data, using data created with two strategies: Random Sampling and Max N-gram.

**Table 10**, **Table 11**, **Table 12**, and **Table 13** display the relative performance of all tasks in Arabic, English, French, and Turkish, respectively, when considering the aforementioned strategies. These results support for our main claim in §5.

**Figure 11** illustrates a compression between two strategies for creating the validation set: proportional and full. In the few-shot setup, the proportional strategy constructs a validation set that is proportionate to the training set (compared to the full training set). For example, we train a model using 10% of the original training set and an equivalent 10% of the original validation set. Conversely, the full strategy involves consistently training the system with a complete validation set, such as 10% of the training set and the entire validation set. The experimental results demonstrate that the inclusion of a full validation set yields only marginal or negligible improvements in system performance. Consequently, it is recommended that future system developers allocate their efforts towards the collection of high-quality in-language datasets, as opposed to focusing on acquiring a full validation set, which has traditionally been done in existing literature. These results support our auxiliary finding **1)** in §5.

**Figure 12** illustrates a comparison between two strategies for creating the training set in a hypothetical setup aimed at developing dialogue systems in multiple languages simultaneously. The **Same** strategy involves training a system using the same set of dialogue patterns in all four languages, utilising a multi-parallel dataset. On the other hand, the **Different** strategy employs different training sets with mutually exclusive dialogue patterns for training the system. For instance, a system trained with 20% of the training data using the **Different** strategy will encounter 80% of the dialogue patterns. The results demonstrate that the **Different** strategies outperform the **Same** strategies in terms of the NLU task. Specifically, covering diverse dialogue patterns yields marginally better performance for NLU. However, this pattern does not hold true for NLG. These results support our auxiliary finding **2)** in §5.

| % of Training Data | ID | | SL | | | DST | | | NLG | | |
|---|---|---|---|---|---|---|---|---|---|---|---|
| | Accuracy | F1 | Precision | Recall | F1 | JGA | Turn Acc. | F1 | BLEU | ROUGE | METEOR |
| Random Sampling | | | | | | | | | | | |
| 1% | 84.7 | 89.8 | 33.4 | 37.5 | 35.3 | 2.7 | 84.9 | 46.9 | 5.5 | 3.6 | 16.5 |
| 5% | 89.5 | 92.7 | 38.3 | 44.2 | 40.9 | 14.1 | 91.9 | 70.3 | 9.5 | 5.9 | 23.8 |
| 10% | 89.2 | 92.8 | 39.8 | 45.0 | 42.3 | 32.0 | 95.2 | 82.5 | 12.4 | 8.4 | 28.1 |
| 50% | 92.4 | 94.7 | 41.5 | 47.4 | 44.3 | 45.6 | 96.6 | 88.2 | 16.0 | 13.4 | 33.4 |
| 100% | 92.7 | 95.0 | 42.4 | 48.5 | 45.2 | 47.9 | 96.9 | 89.4 | 17.9 | 15.0 | 36.0 |
| Max N-gram | | | | | | | | | | | |
| 1% | 86.5 | 91.0 | 33.5 | 38.1 | 35.6 | 6.6 | 87.7 | 58.0 | 6.7 | 5.0 | 18.8 |
| 5% | 90.4 | 93.6 | 39.9 | 45.8 | 42.6 | 23.9 | 94.1 | 78.8 | 10.5 | 10.0 | 24.7 |
| 10% | 91.1 | 94.0 | 40.4 | 46.4 | 43.2 | 33.8 | 95.4 | 83.8 | 11.9 | 11.9 | 27.7 |
| 50% | 92.1 | 94.6 | 43.1 | 48.4 | 45.6 | 46.7 | 96.7 | 88.7 | 16.9 | 14.1 | 34.4 |
| 100% | 92.7 | 95.0 | 42.4 | 48.5 | 45.2 | 47.9 | 96.9 | 89.4 | 17.9 | 15.0 | 36.0 |
| Equal Domain | | | | | | | | | | | |
| 1% | 85.1 | 90.3 | 32.8 | 36.7 | 34.7 | 2.9 | 82.7 | 40.5 | 4.9 | 5.1 | 17.3 |
| 5% | 89.0 | 92.5 | 37.0 | 43.3 | 39.9 | 16.3 | 92.6 | 73.4 | 9.1 | 7.1 | 24.9 |
| 10% | 89.9 | 93.1 | 39.2 | 44.5 | 41.7 | 26.4 | 94.5 | 80.1 | 11.5 | 7.1 | 27.7 |
| 50% | 92.1 | 94.6 | 41.5 | 47.2 | 44.2 | 44.9 | 96.6 | 88.4 | 16.0 | 13.3 | 35.2 |
| 100% | 92.7 | 95.0 | 42.4 | 48.5 | 45.2 | 47.9 | 96.9 | 89.4 | 17.9 | 15.0 | 36.0 |
| Equal Slot | | | | | | | | | | | |
| 1% | 85.1 | 89.6 | 34.1 | 38.9 | 36.4 | 9.5 | 89.7 | 62.3 | 6.0 | 4.1 | 17.4 |
| 5% | 88.6 | 92.1 | 37.5 | 42.8 | 40.0 | 24.2 | 93.9 | 78.1 | 10.5 | 8.6 | 24.7 |
| 10% | 90.0 | 93.3 | 36.0 | 43.5 | 39.4 | 31.6 | 95.2 | 82.5 | 12.3 | 10.4 | 27.2 |
| 50% | 92.4 | 94.9 | 41.3 | 45.9 | 43.6 | 45.2 | 96.6 | 88.1 | 16.2 | 13.6 | 33.3 |
| 100% | 92.7 | 95.0 | 42.4 | 48.5 | 45.2 | 47.9 | 96.9 | 89.4 | 17.9 | 15.0 | 36.0 |
| Max Length | | | | | | | | | | | |
| 1% | 86.9 | 91.2 | 32.9 | 37.3 | 35.0 | 6.3 | 88.0 | 57.1 | 5.8 | 4.7 | 17.3 |
| 5% | 89.6 | 93.2 | 38.7 | 43.8 | 41.1 | 23.7 | 94.1 | 79.4 | 10.2 | 9.0 | 24.9 |
| 10% | 90.7 | 93.8 | 40.2 | 45.8 | 43.8 | 34.0 | 95.6 | 84.1 | 12.0 | 11.0 | 27.7 |
| 50% | 92.5 | 95.0 | 42.6 | 47,2 | 44.8 | 45.5 | 96.6 | 88.4 | 17.2 | 15.0 | 35.2 |
| 100% | 92.7 | 95.0 | 42.4 | 48.5 | 45.2 | 47.9 | 96.9 | 89.4 | 17.9 | 15.0 | 36.0 |

Table 10: The performance of ID, SL, DST, and NLU models trained on an increasing percentage of Arabic data, generated using data creation strategies in §5 *(RQ4)*. The ID and SL models are built upon XLM-R$_{large}$, and the DST and NLG models are based on mT5$_{small}$.

| % of | ID | | SL | | | DST | | | NLG | | |
|---|---|---|---|---|---|---|---|---|---|---|---|
| Training Data | Accuracy | F1 | Precision | Recall | F1 | JGA | Turn Acc. | F1 | BLEU | ROUGE | METEOR |
| | | | | | Random Sampling | | | | | | |
| 1% | 81.6 | 88.3 | 86.3 | 88.3 | 87.3 | 17.0 | 91.1 | 71.5 | 8.4 | 29.7 | 26.9 |
| 5% | 88.8 | 93.4 | 91.2 | 94.2 | 92.6 | 39.8 | 96.1 | 86.1 | 14.8 | 41.2 | 37.5 |
| 10% | 91.0 | 94.9 | 92.1 | 95.1 | 93.5 | 43.2 | 96.5 | 87.5 | 17.0 | 43.7 | 39.7 |
| 50% | 92.7 | 95.8 | 94.3 | 94.9 | 94.6 | 56.4 | 97.7 | 92.8 | 20.4 | 47.3 | 43.0 |
| 100% | 93.2 | 96.1 | 94.6 | 95.7 | 95.1 | 59.8 | 97.9 | 93.5 | 20.8 | 48.4 | 44.1 |
| | | | | | Max N-gram | | | | | | |
| 1% | 84.8 | 90.6 | 87.6 | 89.8 | 88.7 | 17.3 | 92.4 | 73.2 | 11.2 | 35.1 | 31.8 |
| 5% | 89.2 | 93.5 | 92.1 | 93.5 | 92.8 | 41.2 | 96.3 | 87.0 | 16.8 | 43.4 | 39.6 |
| 10% | 90.4 | 94.0 | 92.8 | 94.6 | 93.7 | 47.5 | 96.9 | 89.6 | 18.6 | 45.4 | 41.5 |
| 50% | 92.6 | 94.6 | 94.2 | 94.8 | 94.5 | 57.6 | 97.8 | 93.0 | 20.4 | 47.7 | 43.4 |
| 100% | 93.2 | 96.1 | 94.6 | 95.7 | 95.1 | 59.8 | 97.9 | 93.5 | 20.8 | 48.4 | 44.1 |
| | | | | | Equal Domain | | | | | | |
| 1% | 82.4 | 89.2 | 85.1 | 88.1 | 86.6 | 12.7 | 91.0 | 68.0 | 9.2 | 31.6 | 31.2 |
| 5% | 88.5 | 93.1 | 91.9 | 94.0 | 92.9 | 34.5 | 95.6 | 84.2 | 15.0 | 41.1 | 39.7 |
| 10% | 89.4 | 93.9 | 92.2 | 94.2 | 93.2 | 45.7 | 96.7 | 88.9 | 16.1 | 42.5 | 42.4 |
| 50% | 92.3 | 95.7 | 94.0 | 95.4 | 94.7 | 57.1 | 97.7 | 93.1 | 20.2 | 47.3 | 42.9 |
| 100% | 93.2 | 96.1 | 94.6 | 95.7 | 95.1 | 59.8 | 97.9 | 93.5 | 20.8 | 48.4 | 44.1 |
| | | | | | Equal Slot | | | | | | |
| 1% | 83.2 | 89.3 | 93.9 | 95.8 | 94.8 | 19.2 | 92.3 | 73.3 | 10.0 | 32.8 | 29.4 |
| 5% | 3.9 | 0 | 93.9 | 95.8 | 94.8 | 37.9 | 95.8 | 85.3 | 15.7 | 42.3 | 38.3 |
| 10% | 90.3 | 94.4 | 94.6 | 95.7 | 95.2 | 44.0 | 96.5 | 88.2 | 17.2 | 44.2 | 39.8 |
| 50% | 92.4 | 95.7 | 94.6 | 95.7 | 95.1 | 56.5 | 97.7 | 93.0 | 20.1 | 47.6 | 43.4 |
| 100% | 93.2 | 96.1 | 94.6 | 95.7 | 95.1 | 59.8 | 97.9 | 93.5 | 20.8 | 48.4 | 44.1 |
| | | | | | Max Length | | | | | | |
| 1% | 84.5 | 90.0 | 88.0 | 91.7 | 89.8 | 15.4 | 91.6 | 71.0 | 10.8 | 34.2 | 31.2 |
| 5% | 89.5 | 93.8 | 92.2 | 93.8 | 93.0 | 38.6 | 96.0 | 86.0 | 16.9 | 43.9 | 39.7 |
| 10% | 90.4 | 93.8 | 92.6 | 93.6 | 93.1 | 49.3 | 97.1 | 90.3 | 18.5 | 46.5 | 42.4 |
| 50% | 92.5 | 95.0 | 94.4 | 95.2 | 94.8 | 57.3 | 97.7 | 93.0 | 20.1 | 47.4 | 42.9 |
| 100% | 93.2 | 96.1 | 94.6 | 95.7 | 95.1 | 59.8 | 97.9 | 93.5 | 20.8 | 48.4 | 44.1 |

Table 11: The performance of ID, SL, DST, and NLU models trained on an increasing percentage of English data, generated using data creation strategies in §5 *(RQ4)*. The ID and SL models are built upon XLM-R$_{\texttt{large}}$, and the DST and NLG models are based on mT5$_{\texttt{small}}$.

| % of | ID | | SL | | | DST | | | NLG | | |
|---|---|---|---|---|---|---|---|---|---|---|---|
| Training Data | Accuracy | F1 | Precision | Recall | F1 | JGA | Turn Acc. | F1 | BLEU | ROUGE | METEOR |
| Random Sampling | | | | | | | | | | | |
| 1% | 79.5 | 86.5 | 64.4 | 64.8 | 64.6 | 3.0 | 83.9 | 46.3 | 4.2 | 21.4 | 16.9 |
| 5% | 85.7 | 90.6 | 71.9 | 72.4 | 72.1 | 20.7 | 94.4 | 81.2 | 8.1 | 30.5 | 25.7 |
| 10% | 86.3 | 90.9 | 74.0 | 72.9 | 73.5 | 31.4 | 95.1 | 82.5 | 9.2 | 33.5 | 28.0 |
| 50% | 89.0 | 92.8 | 78.1 | 78.3 | 78.2 | 44.4 | 96.6 | 88.5 | 12.5 | 38.5 | 33.0 |
| 100% | 89.2 | 93.0 | 76.9 | 79.2 | 78.0 | 49.7 | 97.0 | 91.2 | 13.9 | 40.9 | 35.2 |
| Max N-gram | | | | | | | | | | | |
| 1% | 82.3 | 88.4 | 66.7 | 69.1 | 67.9 | 9.7 | 90.3 | 63.6 | 6.0 | 27.2 | 22.0 |
| 5% | 86.8 | 91.2 | 71.4 | 73.6 | 72.5 | 24.2 | 94.4 | 79.5 | 9.5 | 34.5 | 28.7 |
| 10% | 87.1 | 91.6 | 73.6 | 76.7 | 75.1 | 33.5 | 95.5 | 83.6 | 10.4 | 35.6 | 30.0 |
| 50% | 89.1 | 92.8 | 77.6 | 78.5 | 78.1 | 45.9 | 96.7 | 88.9 | 13.4 | 40.1 | 34.4 |
| 100% | 89.2 | 93.0 | 76.9 | 79.2 | 78.0 | 49.7 | 97.0 | 91.2 | 13.9 | 40.9 | 35.2 |
| Equal Domain | | | | | | | | | | | |
| 1% | 78.9 | 86.5 | 63.3 | 67.2 | 65.2 | 5.3 | 86.9 | 52.7 | 4.8 | 24.2 | 22.4 |
| 5% | 85.7 | 90.6 | 72.8 | 72.9 | 72.8 | 21.4 | 93.4 | 76.3 | 6.6 | 28.4 | 27.2 |
| 10% | 86.8 | 91.5 | 75.0 | 74.8 | 74.9 | 31.9 | 95.1 | 82.5 | 8.6 | 32.1 | 30.0 |
| 50% | 88.5 | 92.7 | 77.7 | 78.9 | 78.3 | 45.7 | 96.6 | 88.7 | 12.8 | 38.8 | 34.2 |
| 100% | 89.2 | 93.0 | 76.9 | 79.2 | 78.0 | 49.7 | 97.0 | 91.2 | 13.9 | 40.9 | 35.2 |
| Equal Slot | | | | | | | | | | | |
| 1% | 83.2 | 88.0 | 68.8 | 69.5 | 69.2 | 7.2 | 88.0 | 58.2 | 5.3 | 24.8 | 20.1 |
| 5% | 86.4 | 90.9 | 73.9 | 70.4 | 71.6 | 23.4 | 93.9 | 77.9 | 9.0 | 33.4 | 28.0 |
| 10% | 86.5 | 91.1 | 74.1 | 74.6 | 74.3 | 36.1 | 95.6 | 84.1 | 9.3 | 33.7 | 28.2 |
| 50% | 88.6 | 92.5 | 77.7 | 78.1 | 77.9 | 44.8 | 96.6 | 88.3 | 12.8 | 39.4 | 33.7 |
| 100% | 89.2 | 93.0 | 76.9 | 79.2 | 78.0 | 49.7 | 97.0 | 91.2 | 13.9 | 40.9 | 35.2 |
| Max Length | | | | | | | | | | | |
| 1% | 82.4 | 88.0 | 69.3 | 70.0 | 89.8 | 6.9 | 89.1 | 60.2 | 6.1 | 27.5 | 22.4 |
| 5% | 86.2 | 90.8 | 73.7 | 74.5 | 93.0 | 26.0 | 94.2 | 79.7 | 8.6 | 32.4 | 27.2 |
| 10% | 87.5 | 91.8 | 75.8 | 76.3 | 93.1 | 37.6 | 95.7 | 84.6 | 10.2 | 35.6 | 30.0 |
| 50% | 88.9 | 92.8 | 77.1 | 78.8 | 94.8 | 45.7 | 96.7 | 88.6 | 13.2 | 40.0 | 34.2 |
| 100% | 89.2 | 93.0 | 76.9 | 79.2 | 78.0 | 49.7 | 97.0 | 91.2 | 13.9 | 40.9 | 35.2 |

Table 12: The performance of ID, SL, DST, and NLU models trained on an increasing percentage of French data, generated using data creation strategies in §5 *(RQ4)*. The ID and SL models are built upon XLM-R$_\text{large}$, and the DST and NLG models are based on mT5$_\text{small}$.

| % of | ID | | SL | | | DST | | | NLG | | |
|---|---|---|---|---|---|---|---|---|---|---|---|
| Training Data | Accuracy | F1 | Precision | Recall | F1 | JGA | Turn Acc. | F1 | BLEU | ROUGE | METEOR |
| | | | | | Random Sampling | | | | | | |
| 1% | 79.9 | 86.8 | 74.8 | 76.5 | 75.7 | 3.6 | 85.7 | 48.1 | 7.0 | 27.1 | 24.5 |
| 5% | 88.4 | 92.4 | 81.8 | 83.4 | 82.6 | 27.3 | 94.4 | 81.2 | 13.6 | 39.1 | 34.6 |
| 10% | 88.3 | 92.2 | 84.6 | 84.9 | 84.7 | 35.9 | 95.6 | 85.1 | 17.2 | 44.6 | 39.9 |
| 50% | 91.4 | 94.5 | 87.2 | 85.6 | 86.4 | 48.2 | 96.9 | 89.7 | 21.8 | 50.6 | 45.7 |
| 100% | 92.2 | 95.0 | 86.6 | 87.6 | 87.1 | 52.9 | 97.3 | 91.2 | 24.2 | 53.7 | 48.6 |
| | | | | | Max N-gram | | | | | | |
| 1% | 85.0 | 89.9 | 77.9 | 79.4 | 78.6 | 9.5 | 90.2 | 66.1 | 10.6 | 33.8 | 30.2 |
| 5% | 89.1 | 92.9 | 83.4 | 85.1 | 84.3 | 22.8 | 94.2 | 79.1 | 16.4 | 43.6 | 38.5 |
| 10% | 89.8 | 93.4 | 84.9 | 85.3 | 85.1 | 37.4 | 95.9 | 85.9 | 19.0 | 46.7 | 41.9 |
| 50% | 91.8 | 94.7 | 86.4 | 87.2 | 86.6 | 50.5 | 97.1 | 90.2 | 22.9 | 52.1 | 46.9 |
| 100% | 92.2 | 95.0 | 86.6 | 87.6 | 87.1 | 52.9 | 97.3 | 91.2 | 24.2 | 53.7 | 48.6 |
| | | | | | Equal Domain | | | | | | |
| 1% | 82.7 | 89.0 | 77.9 | 77.3 | 77.6 | 3.8 | 86.4 | 53.6 | 7.8 | 27.7 | 30.5 |
| 5% | 86.9 | 91.6 | 83.5 | 82.6 | 83.0 | 25.6 | 94.0 | 79.8 | 14.6 | 40.5 | 37.8 |
| 10% | 89.8 | 93.4 | 83.5 | 85.0 | 84.2 | 37.5 | 95.7 | 84.9 | 16.0 | 42.8 | 42.2 |
| 50% | 91.5 | 94.5 | 86.0 | 86.4 | 86.2 | 47.7 | 96.9 | 89.7 | 22.5 | 51.5 | 47.4 |
| 100% | 92.2 | 95.0 | 86.6 | 87.6 | 87.1 | 52.9 | 97.3 | 91.2 | 24.2 | 53.7 | 48.6 |
| | | | | | Equal Slot | | | | | | |
| 1% | 83.4 | 89.2 | 80.5 | 80.9 | 80.7 | 7.6 | 89.8 | 62.9 | 8.4 | 29.7 | 26.6 |
| 5% | 88.7 | 92.4 | 83.1 | 83.6 | 83.3 | 30.9 | 95.0 | 82.7 | 14.5 | 41.6 | 36.9 |
| 10% | 89.6 | 93.4 | 83.5 | 84.4 | 84.0 | 36.2 | 95.6 | 85.2 | 17.4 | 45.2 | 40.5 |
| 50% | 91.6 | 94.6 | 86.0 | 86.8 | 86.4 | 49.1 | 97.0 | 89.8 | 22.6 | 51.6 | 46.5 |
| 100% | 92.2 | 95.0 | 86.6 | 87.6 | 87.1 | 52.9 | 97.3 | 91.2 | 24.2 | 53.7 | 48.6 |
| | | | | | Max Length | | | | | | |
| 1% | 85.8 | 90.5 | 76.5 | 77.0 | 76.8 | 10.5 | 89.8 | 64.2 | 10.2 | 34.2 | 30.5 |
| 5% | 89.0 | 92.7 | 83.6 | 84.2 | 83.9 | 30.4 | 95.1 | 83.1 | 15.8 | 42.5 | 37.8 |
| 10% | 89.9 | 93.6 | 84.8 | 84.9 | 84.9 | 39.7 | 96.2 | 86.6 | 18.8 | 46.9 | 42.2 |
| 50% | 91.9 | 94.9 | 85.1 | 86.7 | 85.9 | 50.2 | 97.0 | 90.0 | 23.4 | 52.5 | 47.4 |
| 100% | 92.2 | 95.0 | 86.6 | 87.6 | 87.1 | 52.9 | 97.3 | 91.2 | 24.2 | 53.7 | 48.6 |

Table 13: The performance of ID, SL, DST, and NLU models trained on an increasing percentage of Turkish data, generated using data creation strategies in §5 *(RQ4)*. The ID and SL models are built upon XLM-R$_{\text{large}}$, and the DST and NLG models are based on mT5$_{\text{small}}$.

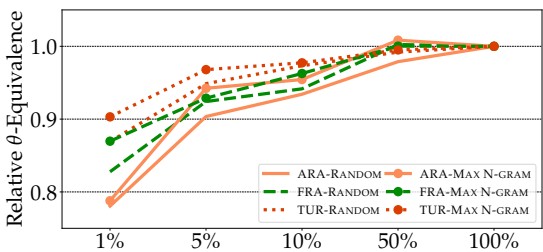

Figure 9: The relative equivalence in performance of SL models trained on an increasing percentage of data in Arabic, French, and Turkish, generated using both the **Random Sampling** and **Max N-gram** strategies. The SL models are built upon $\texttt{XLM-R}_{\texttt{large}}$ and evaluated using the F1 metric.

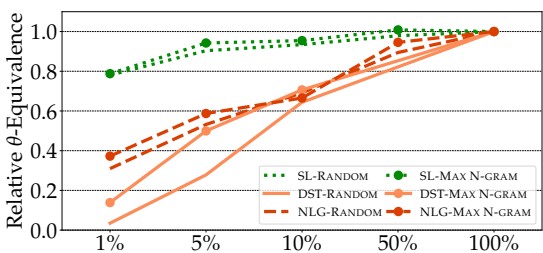

Figure 10: The relative performance of SL and DST models trained on an increasing percentage of Arabic data, generated using both the **Random Sampling** and **Max N-gram** strategies, in comparison to the performance of $P^{\text{ARA}}(\cdot)$. The SL models are built upon $\texttt{XLM-R}_{\texttt{large}}$ and evaluated using the F1 metric, while the DST models are based on $\texttt{mT5}_{\texttt{small}}$ and evaluated using the JGA metric. In addition, we observed performance gains for ID, although the improvements were marginal and not visually represented in the figure.

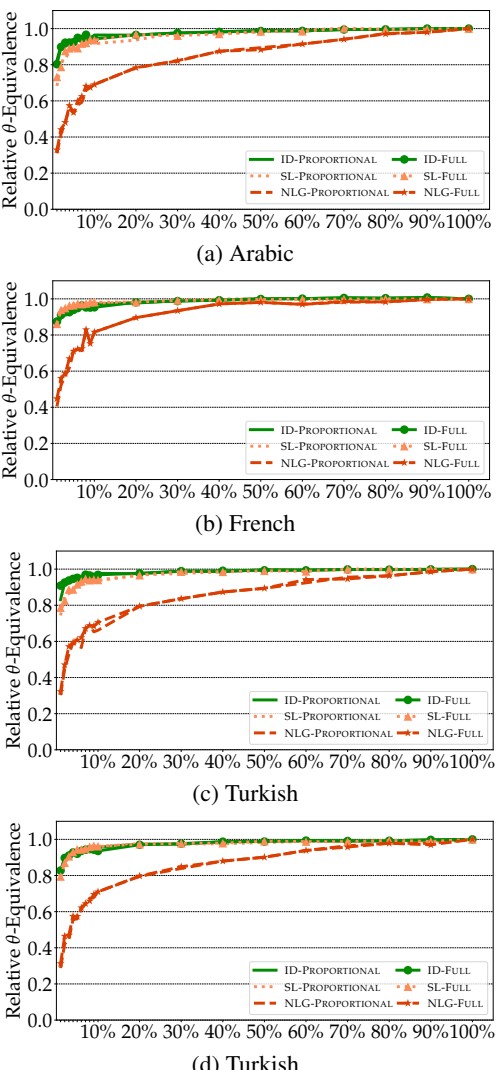

(a) Arabic

(b) French

(c) Turkish

(d) Turkish

Figure 11: Relative performance for ID, SL, and NLG for system trained on increasing percentage of in-language data for **(a)** Arabic, **(b)** English, **(c)** French, and **(d)** Turkish, with respect to fully supervised systems, namely $P^{\text{ARA}}(\cdot)$, $P^{\text{ENG}}(\cdot)$, $P^{\text{FRA}}(\cdot)$, and $P^{\text{TUR}}(\cdot)$, trained on full sized monolingual in-language data. We compared two strategies for creating the validation set: proportional and full. In the few-shot setup, the proportional strategy constructs a validation set that is proportionate to the training set (compared to the full training set). For example, we train a model using 10% of the original training set and an equivalent 10% of the original validation set. Conversely, the full strategy involves consistently training the system with a complete validation set, such as 10% of the training set and the entire validation set. The ID and SL models are built upon $\texttt{XLM-R}_{\texttt{base}}$ and evaluated with Accuracy and F1. The NLG models are based on $\texttt{mT5}_{\texttt{small}}$ and evaluated with BLEU. Due to the significant computational demand, we did not measure the performance of the DST models in these experiments.

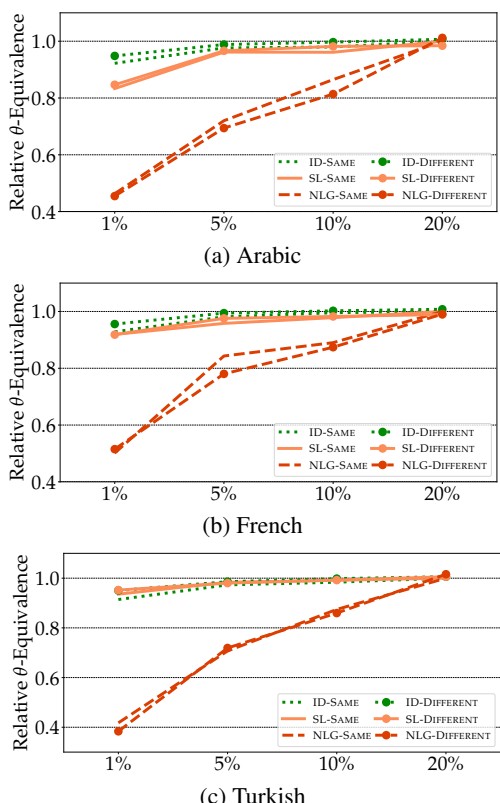

(a) Arabic

(b) French

(c) Turkish

Figure 12: The relative performance of ID, SL, and NLG for systems trained on increasing percentages of multilingual data. Specifically, a system trained on 10% of the training data implies that this system is trained using 10% of the training data for all four languages, simultaneously. We evaluated the trained systems on three languages: **(a)** Arabic, **(b)** French, and **(c)** Turkish, with respect to fully supervised systems, namely $P^{\text{ARA}}(\cdot)$, $P^{\text{FRA}}(\cdot)$, and $P^{\text{TUR}}(\cdot)$, trained on full sized monolingual in-language data. We compare two strategies for creating the training set. Firstly, the **Same** strategy refers to training a system with the same set of dialogue patterns in the four languages. In other words, we use the multi-parallel dataset to train the system. Secondly, the **Different** strategy utilises different training sets with mutually exclusive dialogue patterns to train the system. For example, a system trained with 20% of the training data using the **Different** strategy will encounter 80% of the dialogue patterns. The ID and SL models are built upon XLM-R$_{\text{large}}$ and evaluated with Accuracy and F1. The NLG models are based on mT5$_{\text{small}}$ and evaluated with BLEU. Due to the significant computational demand, we did not measure the performance of the DST models in these experiments.