# OpenReview forum: "A Systematic Study of Performance Disparities in Multilingual Task-Oriented Dialogue Systems"
_EMNLP/2023/Conference — EMNLP 2023 Main_

### Official Review · Reviewer_5jxt · 2023-08-03

**Soundness:** 3

**Excitement:**

4: Strong: This paper deepens the understanding of some phenomenon or lowers the barriers to an existing research direction.

**Missing References:**

While there might not be a specific missing reference, the paper may benefit from offering a more in-depth analysis of related research to distinguish its contributions from others, even though common beliefs surrounding English hegemony already exist in NLP. The following can be one example.
+ Should We Ban English NLP for a Year? (Søgaard, EMNLP 2022)

**Paper Topic And Main Contributions:**

It is a long-standing goal of the NLP community to establish a general system capable of performing effectively across multiple languages. For this goal, the paper focuses on task-oriented dialogue (ToD) systems and empirically analyzes task performance disparities. In addition to standard task performance metrics, the authors propose two primary quantitative measures: absolute and relative equivalence in system performance. Leveraging the latest ToD dataset, MULTI3WoZ, the paper conducts a series of carefully designed experiments under controlled settings, centered around four predefined research questions. The empirical evidence, involving multiple tasks across NLU, DST, and NLG, confirms the presence of adaptation and intrinsic biases in current ToD systems. The role of acquiring high-quality data in a new language is particularly emphasized. The further study provides practical insights for data collection and system development to maximize cost efficiency. Overall, the findings offer valuable guidelines for future research to effectively expand ToD systems across languages.

**Questions For The Authors:**

A. Considering that MULTI3WoZ includes only four languages, should the diversity of languages also be considered one of the required data properties? How to ensure that the findings are not biased by this single dataset and address potential implications on the generalizability of the results (e.g., language family, resource level)?

B. The focus of the paper is on the analysis of task performance disparities, yet RQ 3 and 4 seem to be somewhat off-topic (i.e., data requirements and collection strategy). Please explain the detailed rationale behind the choice of these research questions and how they contribute to the overall research goal of this study.

C. The paper emphasizes the performance disparities between English and other languages, "even with" similar development approaches. However, it is noteworthy that the shared mPLMs themselves already show potential biases in favor of English across languages. Considering this, it is not surprising to assume that the shared mPLMs in similar development lead to the observed performance disparities. Thus, what is the valuable insight from this observation?

D. The paper mentions that the reliability and generalisability of findings are affirmed even with only a single run in the experiments. Could the authors provide more detailed explanations or evidence to support this claim?

**Reasons To Accept:**

- The paper stands as the pioneering systematic analysis of performance disparities in multilingual ToD systems, introducing two fresh quantitative equivalence measures. By conducting experiments on the latest ToD benchmark, MULTI3WoZ, the paper investigates several valuable research questions. The empirical findings bring insights into the challenges and opportunities associated with building ToD systems for different languages.
- The paper conducts comprehensive investigations across multiple types of tasks under controlled settings. Experimental results provide meaningful evidence to support its findings for each research question. The commitment to releasing the data and code fosters transparency and reproducibility.
- The paper provides a good example of utilizing quantitative measures to analyze the performance disparities across languages in ToD systems. By introducing new evaluation metrics, adopting a comprehensive experimental design, and focusing on research-question-oriented paradigms, the study establishes a benchmark for future research, contributing to the advancement of evaluation methodologies, not only in this field but also in multilingual NLP.

**Reasons To Reject:**

+ The paper presents a comprehensive analysis of multiple tasks and models, but its primary reliance on the MULTI3WoZ dataset, encompassing only four languages (Arabic, English, French, and Turkish), raises concerns about the generalizability of the findings to a broader linguistic landscape. While the dataset has its own advantages for the ease of study, the limited scope could potentially bias the conclusions and limit their applicability.
+ The findings of this paper, while insightful, may not be entirely groundbreaking, as similar observations regarding adaptation bias and intrinsic bias have been reported previously. The existence of performance disparities between English and other languages and the need for training data in new languages are widely acknowledged in the NLP community. While the study provides empirical contributions, it does not seem to present entirely novel findings.
+ The clarity of focus in this paper is a concern as it seems to lack a well-defined central theme. Some research questions appear questionable and possibly unrelated to the main objective of analyzing task performance disparities. Additionally, a significant portion of the paper is devoted to non-priority aspects, such as dataset introduction, which might diminish its status as a rigorous research study and instead give the impression of a follow-up analysis blog of the MULTI3WoZ dataset.

**Reproducibility:**

4: Could mostly reproduce the results, but there may be some variation because of sample variance or minor variations in their interpretation of the protocol or method.

**Reviewer Confidence:**

4: Quite sure. I tried to check the important points carefully. It's unlikely, though conceivable, that I missed something that should affect my ratings.

**Typos Grammar Style And Presentation Improvements:**

To enhance the presentation, it is suggested to frontload and clarify the main focus, aligning research questions with the goal of analyzing task performance disparities. Condensing certain parts, such as the dataset introduction, will create space for core analysis and results, emphasizing the primary contributions. Introducing diverse datasets could provide a broader perspective on language performance disparities, enhancing the value of this work.

---

> ### Author Rebuttal · Authors · 2023-08-28
>
> Thank you very much for your helpful review and insightful suggestions. Here are our clarifications to your questions:
>
> ***
>
> **Reason To Reject 1 and Question A: Limited language coverage.**
>
> While this study only covers a small sample of world languages, these languages are from different language families and already exhibit distinct linguistic properties. These properties contain aspects such as word order, grammatical gender, inflectional systems, and verb conjugation. For example, Arabic has complex morphology and a right-to-left script. French has gender and number markers for nouns and adjectives, with agreement extending to determiners. Turkish features a complex system of agglutination and a strict subject-object-verb word order. Our experimental results also confirm that these varied linguistic properties influence the system performance.
>
> Although these languages are not extremely low-resourced, they do belong to the category of under-served languages (Song et al., 2023) in the context of ToD systems because of their high demand (Blasi et al., ACL 2022) and the absence of training data.
>
> Covering a typologically diverse set of languages is a desirable and important property for system development. On the other hand, developing ToD systems in other languages requires new data collection efforts. Our study is timely to provide practical advice on increasing the cost-efficiency of multilingual ToD data collection and thereby increasing the language coverage for future system development. For example, instead of collecting an exhaustive dataset, we might consider gathering only 50% of the dataset to build a DST model achieving a 95% of the JGA. With more available resources for more (lower-resource) languages, we could expand the analysis to a broader linguistic landscape in the future.
>
> References:
>
> Blasi et al., ACL 2022. Systematic Inequalities in Language Technology Performance across the World’s Languages
>
> Song et al., 2023. GlobalBench: A Benchmark for Global Progress in Natural Language Processing
>
> ***
>
> **Reason To Reject 2 and Question C: What is the distinct novelty that this work offers compared to previous research in the field?**
>
> In addition to offering a “pioneering systematic analysis” of performance disparities, focused on the high-impact downstream application of task-oriented dialogue, and empirically confirming the existence of adaptation bias and intrinsic bias, in comparison to prior research, our paper introduces several novel contributions:
>
> 1) We propose measures to quantify the performance disparities (Section 3.2).
> 2) Based on these quantitative measures, our analysis provides practical advice to multilingual ToD system development. For example, Table 1 shows that smaller models outperform their larger counterparts for the NLG tasks. It's worth noting that earlier ToD systems typically employ smaller PLMs for response generation, often without explaining the rationale behind (Section 5 RQ1 and RQ2).
> 3) To the best of our knowledge, we perform the first quantitative analysis of the impact of the scarcity of in-domain, in-language training data on performance disparities (Section 5 RQ3).
> 4) In addition to demonstrating the existence of inherent bias (model bias), our proposed measures enable us to quantify the extent of intrinsic bias. Table 1 and Figure 1 show the extent to which performance drops for each language due to inherent bias in individual PLMs. Furthermore, the experimental results presented in Figure 2 (b) indicate that mPLMs not only perform better in English but also have a faster learning curve in the few-shot scenario, yielding superior relative performance with the same volume of training data compared to other languages.
> 5) Our investigation in response to RQ3 is motivated by challenges encountered in real-world data collection projects, thereby providing practical advice to future multilingual ToD dataset collection. Particularly, we show that the amount of data required depends on the task complexity, with 20% of the in-language data for simpler tasks and more than 50% of the data for more complex tasks such as DST and NLG. Depending on the performance goal of developed systems, future data collection efforts could collect a subset of the full dataset and ultimately increase the language coverage with a fixed budget.
> 6) As a proof of concept, our findings In Section 5 (RQ 4) shows that strategic budget allocation in a multilingual data collection project can enhance model performance without additional costs. We hope that our findings will inspire future research in improving the cost-efficiency in multilingual data collection, thereby broadening language coverage under budget constraints.
>
> ***
>
> **Reason To Reject 3 and Question B: What is the central theme for this paper?**
>
> Our paper focuses on conducting a **quantitative** analysis of performance disparities in the context of multilingual ToD systems, addressing research questions RQ1, RQ2, and RQ3. Following this analysis, we aim to provide **practical advice** on future data collection efforts (RQ3, RQ4), which is a crucial step to minimise these disparities for under-served languages (Song et al., 2023). Our analysis answers an important real-world question: How to balance the amount of data per language and the language coverage under budget constraints? In addition, we show the possibility of enhancing model performance by strategic budget allocation in multilingual data collection, thereby increasing the utility of the language technology with no additional cost (Blasi et al., ACL 2022). As outlined above, the central theme of this paper is: identifying, quantifying, and minimising performance disparity in multilingual ToD systems.
>
> We include a short description of MULTI3WOZ dataset in Section 3.3 because the paper introducing this dataset is under review and not publicly available. In the final version, we will shorten this section.
>
> References:
>
> Blasi et al., ACL 2022. Systematic Inequalities in Language Technology Performance across the World’s Languages
>
> Song et al., 2023. GlobalBench: A Benchmark for Global Progress in Natural Language Processing
>
> ***
>
> **Question D: Can the conclusions be affirmed adequately based on a single run with one random seed?**
>
> In the Limitation section, we acknowledge that our experimental results are derived from a single run due to the high cumulative computational budget of these experiments (> 5,000 GPU hours). Three random runs with varying random seeds will increase the computational budget by more than 10k GPU hours.
>
> Indeed, we did try runs with multiple seeds for less computationally intensive tasks: intent detection and slot labelling. We run parallel experiments for fully supervised models, similar to those reported in Table 1, using three different random seeds. As shown in the following table, the results do not show any real volatility that would impact the main empirical findings and trends in the results.
>
> | Language | ID-Accuracy | ID-F1 | SL-Precision | SL-Recall | SL-F1 |
> |----------|-------------|-------|--------------|-----------|-------|
> | Arabic   | 0.2         | 0.2   | 0.3          | 0.3       | 0.3   |
> | English  | 0.1         | 0.1   | 0.1          | 0.0       | 0.0   |
> | French   | 0.2         | 0.2   | 0.3          | 0.4       | 0.3   |
> | Turkish  | 0.1         | 0.1   | 0.1          | 0.2       | 0.1   |
>
> Table: Standard deviation computed from three experimental runs using varied random seeds. The presented values correspond to each task-metric pair. For instance, the "ID-Accuracy" column represents the standard deviation for accuracy in the intent detection experiment.
>
>
> Nonetheless, we also highlight the consistency of our findings across **all target languages and tasks**. In other words, our findings hold when evaluated through four independent runs for the four target languages. This consistency in outcomes increases our confidence in the reliability and generalisability of our empirical findings.
>
>
>
> ***
>
> **Missing Reference 1: A more in-depth analysis of related research.**
>
> In Section 1, we provide an overview of the background and relevant literature about performance disparities in language technology across languages. Previous studies (e.g., Hu et al., ICML 2020; Lauscher et al., EMNLP 2020; Debnath et al., ACL 2021) have recognised the presence of the performance disparities. However, these studies have not undertaken a comprehensive systematic analysis that covers the ToD domain.
>
> Your suggested stance paper by Søgaard (EMNLP 2022) advocates a seemingly radical but thought-provoking proposal on implementing policies that discourage a predominant focus on English NLP for a period of time. This approach could potentially contribute to the reduction of performance disparities across various languages and make a broader impact with collaborative efforts from the research community
>
> Compared to this work, our paper presents an experimental study on analysing the performance disparities in ToD and offers **practical advice** on multilingual ToD data collection. We will incorporate this reference into the final version of our paper and add a discussion emphasising our contribution compared to related research.
>
> References:
>
> Debnath et al., ACL 2021. Towards more equitable question answering systems: How much more data do you need?
>
> Hu et al., ICML 2020, XTREME: A Massively Multilingual Multi-task Benchmark for Evaluating Cross-lingual Generalisation
>
> Lauscher et al., EMNLP 2020. From Zero to Hero: On the Limitations of Zero-Shot Language Transfer with Multilingual Transformers
>
> Søgaard, EMNLP 2022. Should We Ban English NLP for a Year?
>
> ***
>
> **Presentation Improvement 1.**
>
> Following your suggestions, we will incorporate the following modifications into the final version of our paper:
>
> 1) In Section 5, we will provide a succinct introduction that clarifies the connections between each of our research questions and the central theme of our paper.
> 2) We will condense Section 3.3 by retaining only the essential explanations required to establish the notations used in this paper.
> 3) With the additional space, we will add a discussion highlighting our contribution compared to related research.
> 4) Regarding the dataset, Multi3WOZ is the only dataset suitable for supporting our analysis. In fact, we have reported testing results on a second dataset. Please see the results on GlobalWOZ in Table 1. However, GlobalWOZ does not provide human-created data for system training, which is why we decided to base our empirical study on high-quality human-curated data from Multi3WOZ.
> 5) We leave performing a similar analysis on other NLP applications as a promising future work, and we’ll expand the discussion as suggested.

---

### Official Review · Reviewer_Hf6r · 2023-08-04

**Soundness:** 3

**Excitement:**

4: Strong: This paper deepens the understanding of some phenomenon or lowers the barriers to an existing research direction.

**Paper Topic And Main Contributions:**

Given the importance of multilingual resources for NLP and the need for diaologue systems, this paper introduces a large-scale parallel multilingual task-oriented dialogue, referred to as TOD covering dialog turns over 4 languages (French, Arabic, Turkish, English). The data creation is described and various experiments over this dataset are explained focusing on the performance of the models over each language.The authors describe the notions of "absolute" and "relative equivalence" used to capture performance disparities across languages as well as within each language. The paper claims to emprically prove the existence of the adaptation and intrinsic biases in the ToD systems.

**Questions For The Authors:**

I kept wondering throughout how exactly the smaller datasets were compiled. The answers are provided towards the end of the paper in a satisfactory manner, but I still wondered whether it was too late to introduce them.

**Reasons To Accept:**

The overall theme of the paper is that not all languges benefit equally well from multilingual PLMs, possibly with adverse effects on resource-poor languages. For example, the paper finds that TOD systems trained for Arabic and Turkish with data fully parallel to English TOD display low performance. This is a nice (and experimentaly-supported) finding, though the paper does not fully clarify whether this result is obtained due to a mismatch between data evaluation metrics (BLEU, etc.) and human judgement, the typological difference of these languages, or other reasons. The reason probably lies somewhere in between. In fact, the authors acknowledge the recent research in the field highlighting the fact that automatic evaluation metrics only moderately correspond to human judgement. I agree with the authors that there's need for further research  regarding data evaluation metrics versus human annotations, and it's good that they mention this point in the paper.

**Reasons To Reject:**

I do not have strong reasons to reject this paper, though I felt that the experiments required a full understanding the attachments, which I had expected to quickly skim. It turned out that much of the detail skipped in the current paper is provided in the attachment, which is a paper submitted to a different platform.

It's not clear to me what the authors mean in lines 713-714 concerning the need for finer details at the annotation stage. I understand the authors' point that the authors couldn't delve into the intricacies of cost variations in ML data annotation, but why should we need details at a finer level of granularity to achieve success in different languages? This point probably relates to the data collection methodology explained in the attachment but not in the current paper.

In lines 440 and the following paragraph, zero-shot cross-lingual evaluations are carried out by training an English system using an English dataset. The performance of the system is tested in target languages. It is not clear to me why an English system was trained rather than a multilingual model.

**Reproducibility:**

4: Could mostly reproduce the results, but there may be some variation because of sample variance or minor variations in their interpretation of the protocol or method.

**Reviewer Confidence:**

3: Pretty sure, but there's a chance I missed something. Although I have a good feel for this area in general, I did not carefully check the paper's details, e.g., the math, experimental design, or novelty.

---

> ### Author Rebuttal · Authors · 2023-08-28
>
> Thank you very much for your insightful review. Here are our clarifications to your questions:
>
> ***
>
> **Reason To Reject 1: Some details are in the Appendix.**
>
> Due to the limited space, the Appendix describes:
> 1) our experimental setup (e.g., hyperparameter choices and GPU runtime),
> 2) additional experiential results to show that our findings hold across each target language and each ToD task, where we aimed to summarise all the key findings in the main paper,
> 3) results to support our auxiliary findings, which will also be further expanded in the main paper given one extra page,
> 4) an anonymised version for the Multi3WOZ paper following the Double-Blind Instructions. This will be removed after its formal publication as the Multi3WOZ work will also become publicly available — the actual EMNLP submission is a self-contained submission, and we enclosed the version of the Multi3WOZ paper for interested readers who want to know more about the collection process and the data (as at the time of the paper submission, the Multi3WOZ paper was not publicly available).
>
> ***
>
> **Reason To Reject 2: Confusion in “modelling the annotation process with more fine-grained details”.**
>
> Sorry for the confusion. In Section 5 (RQ 4), our findings have shown, as a proof of concept, that strategic budget allocation in a multilingual data collection project can enhance model performance without additional costs. In our analysis, we abstract away many “fine-grained details” in a real-word data collection project. For example, we assume that the annotation cost for all data entries in all languages are identical. However, human annotators are usually paid based on the amount of time they devote. Consequently, annotating each data entry may cost differently. For example, translating a long dialogue may cost more than a short one simply due to the increased time consumption. We will make these ‘fine lines’ clearer in the paper.
>
> We hope that our findings could inspire future work to delve into more sophisticated annotation strategies and conduct a “more fine-grained” analysis. In other words, we encourage future analysis to consider these finer details of real-world data collection. It does not necessarily mean that a multilingual data collection project needs to collect annotations at a finer level of granularity to be successful. We agree that this sentence (Lines 713 – 714 in the Limitation Section) is confusing. We will clarify it in the final version.
>
> ***
>
> **Reason To Reject 3: Why report zero-shot performance relative to English for all target languages?**
>
> In the zero-shot cross-lingual transfer scenario, where task-specific data for target languages is absent, we exploit abundant task-specific data from resource-rich source languages (in our study, English) to make predictions in target languages (Arabic, French, and Turkish). To elaborate, we finetune **multilingual PLMs** with English in-domain dialogue data and evaluate these models with testing data in each target language. This well-established paradigm has been widely used in previous research, as exemplified by the work of Lauscher et al. (EMNLP 2020) and the work by Ansell et al. (ACL 2023), among others.
>
> References:
>
> Ansell et al., ACL 2023. Distilling Efficient Language-Specific Models for Cross-Lingual Transfer
>
> Lauscher et al., EMNLP 2020. From Zero to Hero: On the Limitations of Zero-Shot Language Transfer with Multilingual Transformers
>
> ***
>
> **Question 1: Clarification on the procedure of sampling few-shot data for RQ3.**
>
> In line 535 and the caption of Figure 2, we will add a sentence clarifying that we randomly select a subset (e.g., 20%) from the complete training dataset for few-shot training, and we’ll also release the subsets for reproducibility.

---

### Official Review · Reviewer_Pe2h · 2023-08-05

**Soundness:** 4

**Excitement:**

3: Ambivalent: It has merits (e.g., it reports state-of-the-art results, the idea is nice), but there are key weaknesses (e.g., it describes incremental work), and it can significantly benefit from another round of revision. However, I won't object to accepting it if my co-reviewers champion it.

**Paper Topic And Main Contributions:**

This paper conducts an exhaustive and systematic study for empirically evaluating task performance disparities between multilingual task-oriented dialogue (ToD) systems. They define new quantitative measures of absolute and relative equivalence to quantity the disparities within and across languages. Overall, they answer four research questions basing their conclusions on a number of factors like the nature of the task, the underlying pre-trained model, the source and target languages, and the amount of ToD annotated data. Overall, their work aims at aiding in better collection of ToD data which in turn can boost better development of multilingual ToD models.

**Reasons To Accept:**

- Systematic Study of Multilingual ToD systems: This work provides a good systematic and experimentally sound study for evaluating the disparities across ToD models within languages and across languages. The experimentation is well-backed with results and the paper yields various insightful conclusions.

- Well-written work: This paper is pretty well-written with technically defining difficult concepts. It’s easy to read and follow. Tables and figures have elaborate captions which make them easy to understand.

**Reasons To Reject:**

- Less languages studied: Although this is an issue of the underlying dataset, but the insights may not be generalizable to all languages since it’s been studied only for 3 languages overall, none of which are low-resource in nature

**Reproducibility:**

4: Could mostly reproduce the results, but there may be some variation because of sample variance or minor variations in their interpretation of the protocol or method.

**Reviewer Confidence:**

3: Pretty sure, but there's a chance I missed something. Although I have a good feel for this area in general, I did not carefully check the paper's details, e.g., the math, experimental design, or novelty.

**Typos Grammar Style And Presentation Improvements:**

- Section 5: It would be good to include the research questions included in the first section here for better readability.

---

> ### Author Rebuttal · Authors · 2023-08-28
>
> Thank you very much for your encouraging review. Here, our clarifications to your questions and concerns:
>
> ***
>
> **Reason To Reject 1: Limited language coverage.**
>
> While we agree that the present study only covers a small sample of world languages, these languages are from different language families and already exhibit distinct linguistic properties. These properties contain aspects such as word order, grammatical gender, inflectional systems, and verb conjugation. For example, Arabic has complex morphology and a right-to-left script. French has gender and number markers for nouns and adjectives, with agreement extending to determiners. Turkish features a complex system of agglutination and a strict subject-object-verb word order. Our experimental results also confirm that these varied linguistic properties influence the system performance.
>
> Although these languages are not extremely low-resourced, they do belong to the category of under-served languages (Song et al., 2023) in the context of ToD systems because of their high demand (Blasi et al., ACL 2022) and the absence of training data.
>
> Developing ToD systems in other languages requires new data collection efforts. Our study is timely to provide **practical advice** on increasing the cost-efficiency of multilingual ToD data collection and thereby increasing the language coverage for future system development. For example, instead of collecting an exhaustive dataset, we might consider gathering only 50% of the dataset to build a DST model achieving a 95% of the JGA. With more available resources for more (lower-resource) languages in the future, we could expect to expand the analysis to a broader linguistic landscape in the future.
>
> References:
>
> Blasi et al., ACL 2022. Systematic Inequalities in Language Technology Performance across the World’s Languages
>
> Song et al., 2023. GlobalBench: A Benchmark for Global Progress in Natural Language Processing
>
> ***
>
> **Presentation Improvement 1: Including RQs in Section 5.**
>
> We will expand the introductory paragraph of Section 5 and clarify how each research question is related to the subsequent analysis following your suggestion.

---

### Meta-Review · Area_Chair_ghvy · 2023-09-26

**Recommendation:** 4

**Metareview:**

This paper provides an empirical analysis of task performance disparities between multilingual task-oriented dialogue (ToD) systems, demonstrating that these disparities depend on the nature of the task at hand, the underlying pre-trained language model, the target language, and the amount of annotated data.  The findings offer valuable guidelines for future research to effectively expand ToD systems across languages.  Two of the reviewers express a concern that only four languages were covered, and that therefore the findings may not generalise to other languages.  The authors addressed this to some extent in their rebuttal by pointing out that the four languages were from different families, emphasising their linguistic variation.

---

### Decision · Program_Chairs · 2023-10-07

**Decision:**

Accept-Main

**Comment:**

This paper provides an empirical analysis of task performance disparities between multilingual task-oriented dialogue (ToD) systems, demonstrating that these disparities depend on the nature of the task at hand, the underlying pre-trained language model, the target language, and the amount of annotated data.  The findings offer valuable guidelines for future research to effectively expand ToD systems across languages.  Two of the reviewers express a concern that only four languages were covered, and that therefore the findings may not generalise to other languages.  The authors addressed this to some extent in their rebuttal by pointing out that the four languages were from different families, emphasising their linguistic variation.